# A deep learning framework identifies dimensional representations of Alzheimer's Disease from brain structure

Zhijian Yang [1,2,100], Ilya M. Nasrallah [1,3,100], Haochang Shou [1,4], Junhao Wen [1,3], Jimit Doshi[1,3], Mohamad Habes[1,5], Guray Erus[1,3], Ahmed Abdulkadir [1,3], Susan M. Resnick[6], Marilyn S. Albert[7], Paul Maruff[8], Jurgen Fripp[9], John C. Morris[10], David A. Wolk[1,11,12], Christos Davatzikos [1,3 ✉], iSTAGING Consortium*, Baltimore Longitudinal Study of Aging (BLSA)* & Alzheimer's Disease Neuroimaging Initiative (ADNI)*

Heterogeneity of brain diseases is a challenge for precision diagnosis/prognosis. We describe and validate Smile-GAN (SeMI-supervised cLustEring-Generative Adversarial Network), a semi-supervised deep-clustering method, which examines neuroanatomical heterogeneity contrasted against normal brain structure, to identify disease subtypes through neuroimaging signatures. When applied to regional volumes derived from T1-weighted MRI (two studies; 2,832 participants; 8,146 scans) including cognitively normal individuals and those with cognitive impairment and dementia, Smile-GAN identified four patterns or axes of neurodegeneration. Applying this framework to longitudinal data revealed two distinct progression pathways. Measures of expression of these patterns predicted the pathway and rate of future neurodegeneration. Pattern expression offered complementary performance to amyloid/tau in predicting clinical progression. These deep-learning derived biomarkers offer potential for precision diagnostics and targeted clinical trial recruitment.

[1] Center for Biomedical Image Computing and Analytics, University of Pennsylvania, Philadelphia, PA, USA. [2] Graduate Group in Applied Mathematics and Computational Science, University of Pennsylvania, Philadelphia, PA, USA. [3] Department of Radiology, University of Pennsylvania, Philadelphia, PA, USA. [4] Department of Biostatistics, Epidemiology and Informatics, University of Pennsylvania, Philadelphia, PA, USA. [5] Neuroimage Analytics Laboratory (NAL) and the Biggs Institute Neuroimaging Core (BINC), Glenn Biggs Institute for Alzheimer's & Neurodegenerative Diseases, University of Texas Health Science Center San Antonio (UTHSCSA), San Antonio, TX, USA. [6] Laboratory of Behavioral Neuroscience, National Institute on Aging, Baltimore, MD, USA. [7] Department of Neurology, Johns Hopkins University School of Medicine, Baltimore, MD, USA. [8] Florey Institute of Neuroscience and Mental Health, University of Melbourne, Melbourne, VIC, Australia. [9] CSIRO Health and Biosecurity, Australian e-Health Research Centre CSIRO, Brisbane, Queensland, Australia. [10] Knight Alzheimer Disease Research Center, Washington University in St. Louis, St. Louis, MO, USA. [11] Alzheimer's Disease Research Center, University of Pennsylvania, Philadelphia, PA, USA. [12] Department of Neurology, University of Pennsylvania, Philadelphia, PA, USA. [100] These authors contributed equally: Zhijian Yang, Ilya M. Nasrallah. *Lists of authors and their affiliations appear at the end of the paper. ✉email: Christos.Davatzikos@pennmedicine.upenn.edu

Neurologic and neuropsychiatric diseases and disorders are often very heterogeneous in their neuroimaging and clinical phenotypes. Artificial intelligence methods, especially deep-learning approaches, have made a notable leap in medical imaging applications[1] and have shown great promise in deriving individualized neuroimaging signatures of anatomy, function and pathology that offer diagnostic and prognostic value[2]. However, only recently have deep-learning approaches been developed that allow investigation of disease heterogeneity through identification of common but distinct disease subtypes which might have different prognosis, progression patterns, and response to treatments. Toward this goal, a semi-supervised deep-learning paradigm is presented herein (Fig. 1), referred to as Smile-GAN (SeMI-supervised cLustEring via Generative Adversarial Network). Smile-GAN models disease effects via sparse transformations of normal measures, leveraging a GAN that is trained to synthesize transformations producing realistic measures that are hard to distinguish from those derived from real patient data. Estimated latent variables capture phenotypical subtypes, modulating this synthesis in an inverse-consistent formulation which ensures that subtype membership can be reliably estimated from respective biomarker signatures.

Although Smile-GAN is a general methodology, herein it is tested on identifying the heterogeneity in cerebral neuroanatomy —specifically heterogeneity of atrophy as measured by decreases in volumes of gray matter and white matter regions of interest and increases in ventricle volumes—found across a spectrum from early cognitive impairment to dementia among 8146 scans from 2832 individuals across 2 longitudinal cohorts (ADNI, the Alzheimer's Disease Neuroimaging Initiative, and BLSA, the Baltimore Longitudinal Study of Aging[3,4]) with previously harmonized neuroimaging (via the iSTAGING consortium)[5]. Alzheimer's disease (AD) is the most common neurodegenerative disease, affecting millions across the globe[6], and accounts for the majority of cognitive decline in our study sample. The hallmark pathology of AD includes the presence of ß-amyloid neuritic plaques and tau protein-containing neurofibrillary tangles, which contribute to the characteristic neurodegeneration measured on magnetic resonance imaging (MRI). While diagnostic criteria have traditionally focused on the clinical syndrome, typically a predominately amnestic phenotype for AD and a pre-dementia phase called Mild Cognitive Impairment (MCI), recently there has been increasing effort to define AD biologically based on the presence of biomarkers for amyloid deposition (A), tau deposition (T), and neurodegeneration (N), each characterized typically dichotomously as either absent (−) or present (+) and, thus, defining the AT(N) framework[7]. While useful, such binary

characterizations poorly capture biomarker heterogeneity, such as known variability in AD topography or effects of common copathologies, including vascular disease and other comorbid neurodegenerative processes that might affect the 'N' dimension in distinct ways. This variability, along with patient resilience to neuropathology, plays an important role in the ultimate expression of cognitive decline in the individual and is therefore critical to understand when moving beyond group effects of disease to personalized diagnostics. Further, by more clearly identifying typical patterns and severity of neurodegeneration, including patterns more suggestive of underlying AD, such methods may allow improved selection of participants for clinical trials.

Several MRI biomarkers have been used to quantify neurodegeneration in AD. One of the most common is hippocampal volume[8]; hippocampal atrophy is characteristic feature of typical AD. However, as for other single region-of-interest (ROI) markers, it is neither specific for AD, nor does it capture a complex atrophy pattern across the brain that is relevant to the overall phenotype. Composite measures sensitive to the typical temporoparietal atrophy seen in AD, including various regional volumetric signatures[9] or machine learning metrics like SPARE-AD[10,11], provide alternative measures of neurodegeneration that also capture relevant changes across multiple brain regions. These methods provide monolithic signatures of AD-like neurodegeneration with high sensitivity and specificity, but do not elucidate the heterogeneity of neurodegenerative patterns found in AD and its preclinical stages, nor do they attempt to relate such heterogeneity with comorbid pathologies. Recently, novel data-driven methods that leverage large neuroimaging datasets and novel machine learning methodology have emerged to identify patterns of cerebral atrophy in AD and other neurodegenerative diseases.

Clustering methods have been used to identify cross-sectional or temporal heterogeneity in patients[12–16]. Zhang et al.[15] used the Bayesian Latent Dirichlet Allocation (LDA) model to identify latent atrophy patterns from voxel-wise gray matter (GM) density maps derived from structural MRI. Young et al.[12] proposed to uncover temporal and phenotypic heterogeneity by inferring both subtypes and stages. However, these approaches derive clusters only based on patients and hence may identify clusters or patterns partially incorporating disease-irrelevant confounding factors that influence inter-individual brain variations. The semi-supervised method proposed here aims to overcome this limitation by effectively clustering differences between cognitively normal (CN) individuals and patients, thereby focusing on neuroanatomical heterogeneity of pathologic processes rather than heterogeneity that might be caused by a variety of confounding factors[17,18]. Generative adversarial networks (GAN)[19] are well-known for

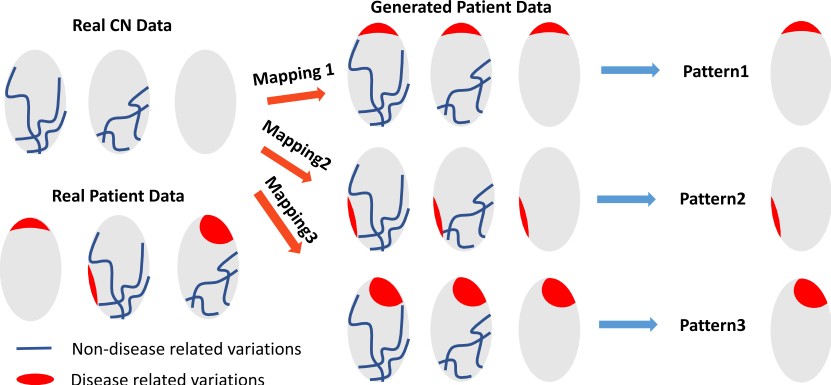

**Fig. 1 Conceptual overview of Smile-GAN.** Blue lines represent non-disease-related variations observed In both normal control (CN) and patient groups. Red regions represent disease effects which only exist among patient groups. Smile-GAN finds neuroanatomical pattern types by means of clustering transformations from CN data to patient data.

learning and modeling complex distributions using a competition between two neural networks. Herein, we used GANs to synthesize exceptionally realistic regional volumes derived from imaging data to model disease effects and perform semi-supervised clustering.

Building on GAN-based models[20–23], the Smile-GAN method captures different disease-related neuroanatomical patterns by generating realistic ROI volume data through transformation of ROI-based neuroanatomical data of CN individuals. Via inverse-consistent latent variables, this synthesis is guided by disease-related neuroanatomical subtypes, which are estimated from the data. Moreover, we extensively validate this method using simulated data as well as synthesized patterns of brain atrophy. Smile-GAN does not directly generate predictions of disease stage but quantifies the degree of expression of captured patterns. Mixed pathologies and staging can be captured via post-hoc, second stage analysis of the expression of derived patterns or combinations thereof.

We hypothesized that Smile-GAN, when trained on a sample enriched for AD, would identify common patterns of neurodegeneration seen in patients along the AD pathway. We discovered 4 reproducible neuroanatomical patterns of atrophy across the spectrum of cognitive decline and developed ways to quantify the level of expression of each of these patterns in any individual, thereby arriving at a 4-dimensional system capturing major patterns of heterogeneity of the 'N' dimension in the AT(N) system. Further, by measuring longitudinal trajectories within this coordinate system, we identified two distinct progression pathways which imply variability in the presence of copathologies and/or heterogeneity of AD pathological processes. We identify baseline patterns with predictive abilities for future neurodegenerative and clinical trajectories for individual participants.

## Results

**Validation of Smile-GAN model on synthetic and semi-synthetic dataset**. Experiments on a synthetic dataset (Supplementary Method 1.3.1) verified the ability of the model to capture heterogeneous disease-related variations while not being confounded by non-disease-related variation. Mapping functions captured all regions with simulated atrophy along each direction while almost perfectly avoiding all regions with much stronger simulated non-disease-related variations. (Supplementary Fig. 1(A)). Experiments on the semi-synthetic dataset, derived from real MRI ROI data but with artificial brain atrophy in selected ROIs (Supplementary Method 1.3.2), further validated the ability of the model to avoid non-disease-related variability under more realistic scenarios. Moreover, the performance of the model was shown to be superior to other state-of-the-artsemi-supervised clustering methods and traditional clustering methods in detecting simulated pattern types even with very small and variable atrophy rates. (See Supplementary Table 4)

**Four patterns of neurodegeneration**. Trained on baseline data of ADNI2/GO participants and validated through the permutation test (Supplementary Result 2.4), Smile-GAN identified four significantly reproducible and disease-related patterns of brain atrophy in cognitively impaired participants. The four-pattern types were found to be reproducible using a holdout cross-validation experiment. Figure 2b plots estimated probabilities of each participant belonging to each pattern in a diamond plot, with each participant colored based on the dominant pattern (a diamond plot is sufficient for visualization of these 4 patterns, because the pattern probabilities sum up to 1 and no participant has both P1 and P4 probabilities >0). Participants with different patterns show distinct atrophy signatures compared to CN, which

are shown by voxel-based group comparison results between the CN group and the groups of participants segregated based on the pattern with highest probability (Fig. 2a). From these results, we can visually interpret the four imaging patterns as: (i) P1, preserved brain volume, exhibits no significant atrophy across the brain compared to CN; (ii) P2, mild diffuse atrophy, with widespread mild cortical atrophy without pronounced medial temporal lobe atrophy; (iii) P3, focal medial temporal lobe atrophy, showing localized atrophy in the hippocampus and the anterior-medial temporal cortex with relative sparing elsewhere; (iv) P4, advanced atrophy, displaying severe atrophy over the whole brain including severe temporal lobe atrophy. These four patterns were highly reproducible when we trained the model on participants from various, independent AD studies (Supplementary Fig. 2), providing further evidence that these are conserved patterns among studies of AD. Moreover, they were reproduced when we trained the model using only participants with positive ß-amyloid (Abeta) status (Supplementary Fig. 3), indicating that these four-pattern probabilities also capture common variation observed among participants who show evidence of AD-related neuro-pathological change. The patterns segregate participants that may demonstrate pathologically identified subtypes of AD[24], as determined by imaging biomarkers[25]. The P3 pattern includes those who may have Limbic Predominant neuropathology and P2 includes those who may have Hippocampal Sparing neuropathology, while mixed P2–P3 may be more typical AD (Supplementary Fig. 5).

**Two progression pathways**. Figure 2c reveals evolution of pattern probabilities over time in the study subsample with longitudinal data. Participants with dominant P1 features at baseline may express increasing probability of P2 or P3 in the short term followed by later expression of the P4 pattern. Participants with dominant P2 or P3 expression at baseline show variable minor expression of the other pattern (other P3/P2 pattern probability range from 0 to 0.5). Both P2 and P3 participants have increasing P4 probability at later time points, but do not develop significant expression of the other P3 or P2 pattern, respectively. Participants who initially had the highest probability of P4 only show stronger expression of P4 over time. From these results, we conclude that P1-2-4 and P1-3-4 are two general MRI progression pathways of neurodegeneration. Figure 2d displays detailed progression paths of some representative participants over time in the pattern-dimension system. These examples demonstrate that despite following similar progression pathways, participants may have difference in pattern purity and progression speed. More specifically, though participants denoted in purple color both show higher P3 probability than P2 during progression process, the solid line is closer to the P2 triangle, showing that this participant has a relatively stronger expression of P2. Also, participants represented by dashed lines progress from P1 to P4 within 5 years (time not shown in the plot), while participants denoted by solid lines take more than 10 years to progress from P1 to P4.

**Amyloid/tau/pattern/diagnosis**. Most of CN participants had negative Abeta status (A−) and express P1 (Fig. 3a). P1 also included the largest number of cognitively impaired but non-demented participants, classified in BLSA/ADNI as MCI, with disproportionately amyloid negative status compared to the other three patterns. There were a comparable amount of MCI/Dementia participants with P2 and P3 (144 and 178) and they had similar distributions in amyloid status, predominantly amyloid positive (66.9% and 72.1%). P4 participants were mostly amyloid positive (84.0%) and were rarely CN (3.3%). Pattern membership can be used to classify participants based on the

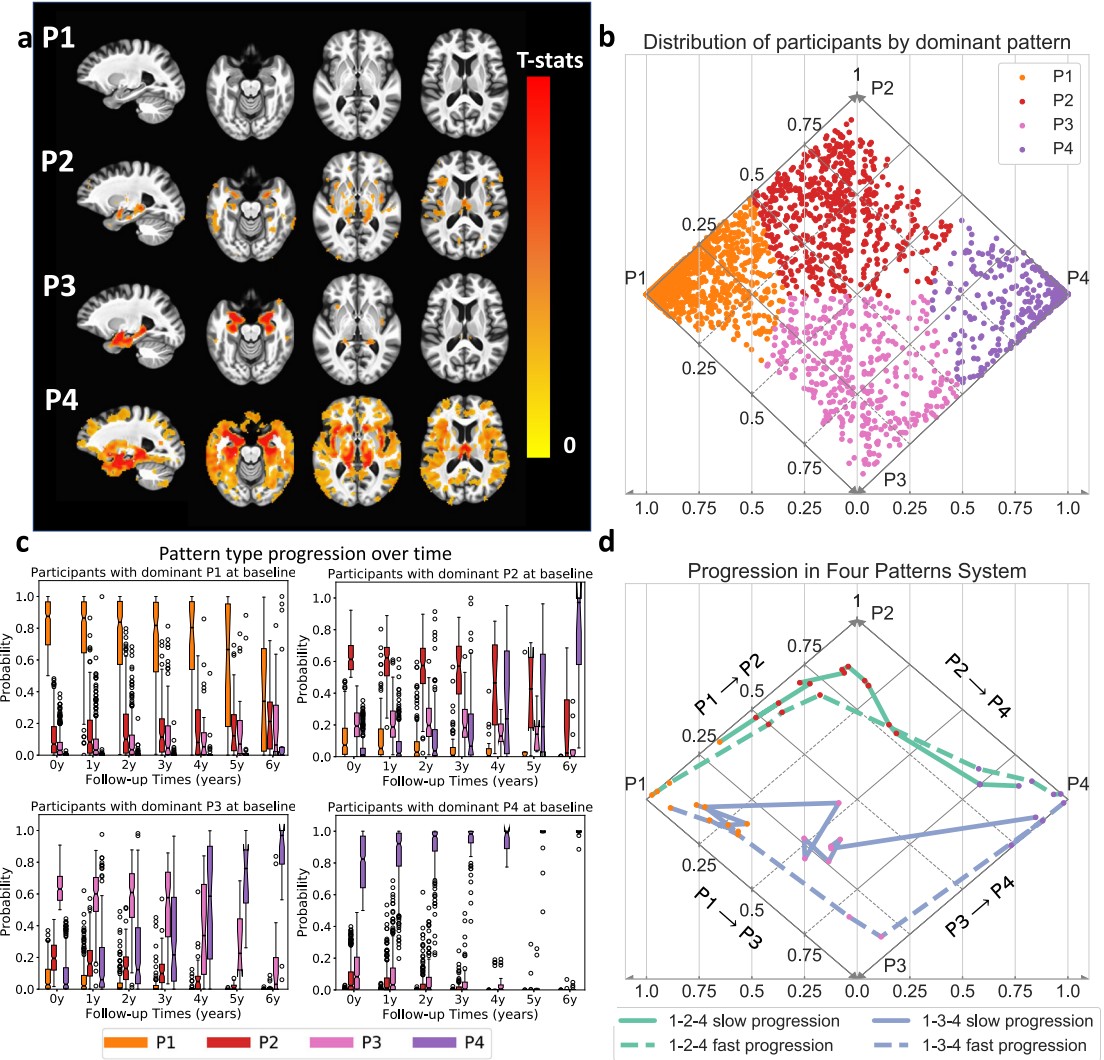

**Fig. 2 Characterization of four atrophy patterns (P1–P4) and two progression pathways of neurodegeneration.** (Data from 899 ADNI2/GO participants in discovery set (**a**) and all 2832 ADNI/BLSA participants (**b**-**d**)). **a** Voxel-wise statistical comparison (one-sided *t*-test) between CN and participants predominantly belonging to each of the four patterns. False discovery rate (FDR) correction for multiple comparisons with *p*-value threshold of 0.05 was applied. **b** Visualization of participants' expression of four patterns in a diamond plot. Pseudo-probabilities of belonging to each pattern reflect levels of expression (i.e., presence) of respective patterns and probabilistic subtype memberships. Horizontal axis indicates p1 and p4 probabilities and diagonal axes reveal p2 (solid lines) and p3 (dashed lines) probabilities. Since participants never have both P1 and P4 > 0, all observed pattern combinations can be represented in this diamond plot. Dots for individual participants are color coded by the dominant pattern. **c** Box and whisker plots of expression of the four patterns over time for each baseline pattern group. (center line, median; box limits, upper and lower quartiles; whiskers, 1.5× interquartile range; points, outliers). **d** Progression paths of four representative participants. Dashed lines show participants reaching P4 from P1 within 5 years and solid lines show those who take more than 10 years to reach P4 from P1. Source data are provided as a Source Data file.

AT(N) criteria, providing insight into the stage of the disease, resilience, and presence of copathology. Those placed along the AD continuum are further subgrouped into early neurodegeneration (P3), advanced (P4) neurodegeneration, or a P2 group of mild diffuse atrophy that may be classified as N− or N+ by other quantification methods. In Fig. 3b, participants are grouped as normal, as falling along the typical AD continuum, as AD with dominant copathology or as suspected non-AD pathology (SNAP) based on patterns and Abeta/phospho-tau (pTau) status. A+T+ participants tend to have more severe neurodegeneration than A+T- participants, as expected. Using pattern membership as a classification of (N) modestly increases the number of classes (from 8 to 16) but provides important severity and prognostic information.

**MRI and clinical characteristics**. Statistical comparisons of MRI and clinical characteristics were conducted among A+ cognitively impaired participants with different dominant patterns (Supplementary Table 6). Relative to P2/P3, P4 and P1 participants showed significantly higher and lower WML volume, respectively, (median (1st–3rd quartile) 50.6 (39.1–66.1) mm³ and 34.1 (25.9–49.6) mm³, *p* < 0.001), but there was no significant difference between P2 and P3 (median 46.3 (30.2–59.2) mm³ and 45.1 (33.6–57.2) mm³, *p* = 0.86). P3 and P4 participants showed significantly lower hippocampal volume relative to total brain volume (median percentage 0.54 (0.51–0.56)% and 0.52 (0.48–0.56)%, respectively, for P3 and P4, versus 0.61 (0.58–0.64)% for P1, both *p* < 0.001). Certain features were the highest in P3 participants: ApoE ε4 allele carrier rate (78%), tTau

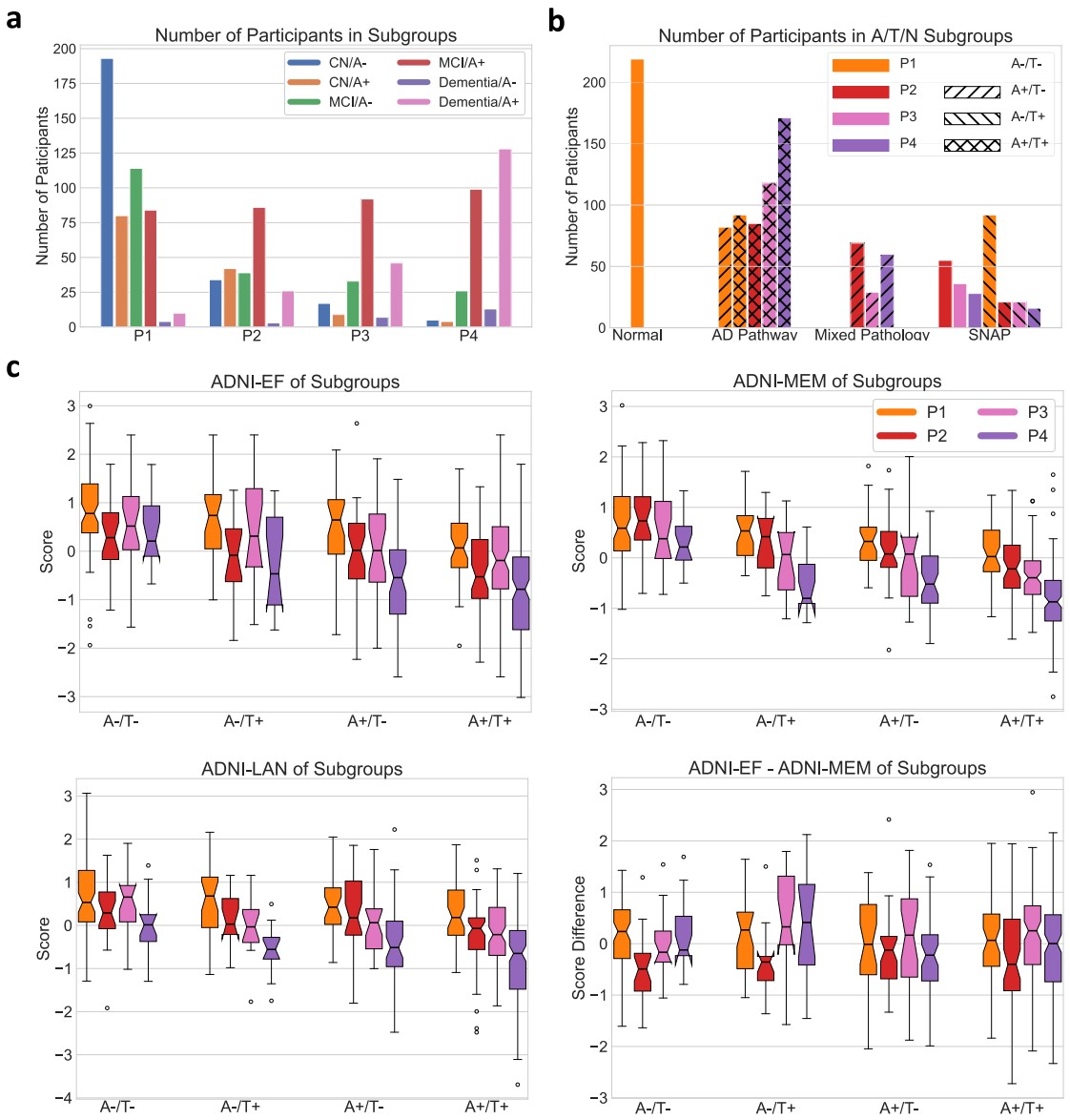

**Fig. 3 Participants grouping and cognitive performance of subgroups.** (Data from 1194 ADNI participants with Abeta/pTau measures) **a** Number of participants grouped by diagnosis, amyloid status, and pattern. **b** AT(N) categorization based on participants' patterns and CSF Abeta/pTau status. Based on patterns, N is classified as normal (P1), not typical of AD (P2), or characteristic of AD (P3/P4). **c** Box and whisker plots of cognitive performance of MCI/Dementia participants by pattern. (A: Abeta; T: pTau) (center line, median; box limits, upper and lower quartiles; whiskers, 1.5× interquartile range; points, outliers). Source data are provided as a Source Data file.

(341.1 (267.9–446.7)) and pTau (34.9(26.1–46.4)) levels. Participants with different patterns were significantly different in cognitive test scores, with important differences based on A/T status (Fig. 3c and Supplementary Table 6). Regardless of A/T status, P1 participants had much better performance across cognitive domains while P4 participants had the poorest performance. P2 participants showed worse performance in executive function than P3 participants but had better function in memory. ADNI-EF and ADNI-MEM were significantly different between P2 and P3 participants ($p = 0.039$ for A−T−, $p = 0.003$ for A−/T+ and $p < 0.001$ for A+/T+). Investigation of special subgroups showed additional features of disease. Within A+P3 participants, CN and impaired groups had similar tTau ($p = 0.33$) and pTau ($p = 0.48$) levels, suggesting comparable AD pathologic change. However, A+P3 CN participants had significantly longer education ($p = 0.001$), higher hippocampal volumes ($p = 0.018$), and somewhat less expression of the P3 pattern probabilities

($p = 0.048$) compared to A+P3 impaired participants, suggesting that higher cognitive reserve and less neurodegeneration may account for the preservation of cognitive function in this relatively small group ($n = 19$, Supplementary Table 7). A−T−P1 and A+T−P1 participants with MCI/Dementia were not significantly different in cognitive test scores or hippocampal volume (Supplementary Table 8), but A+T+P1 participants did show significantly worse cognitive performance (Fig. 2c) along with greater atrophy in hippocampus ($p = 0.011$) and significantly lower P1-probability ($p < 0.001$), suggesting early adverse effects of T likely related to underlying AD pathology even without much neurodegeneration present.

**Longitudinal progression of pattern types.** Cumulative incidence curves in Fig. 4a show that P1 participants at baseline are more likely to progress to P2 than to P3, and that participants with P3 at baseline have a higher chance to progress to P4 than

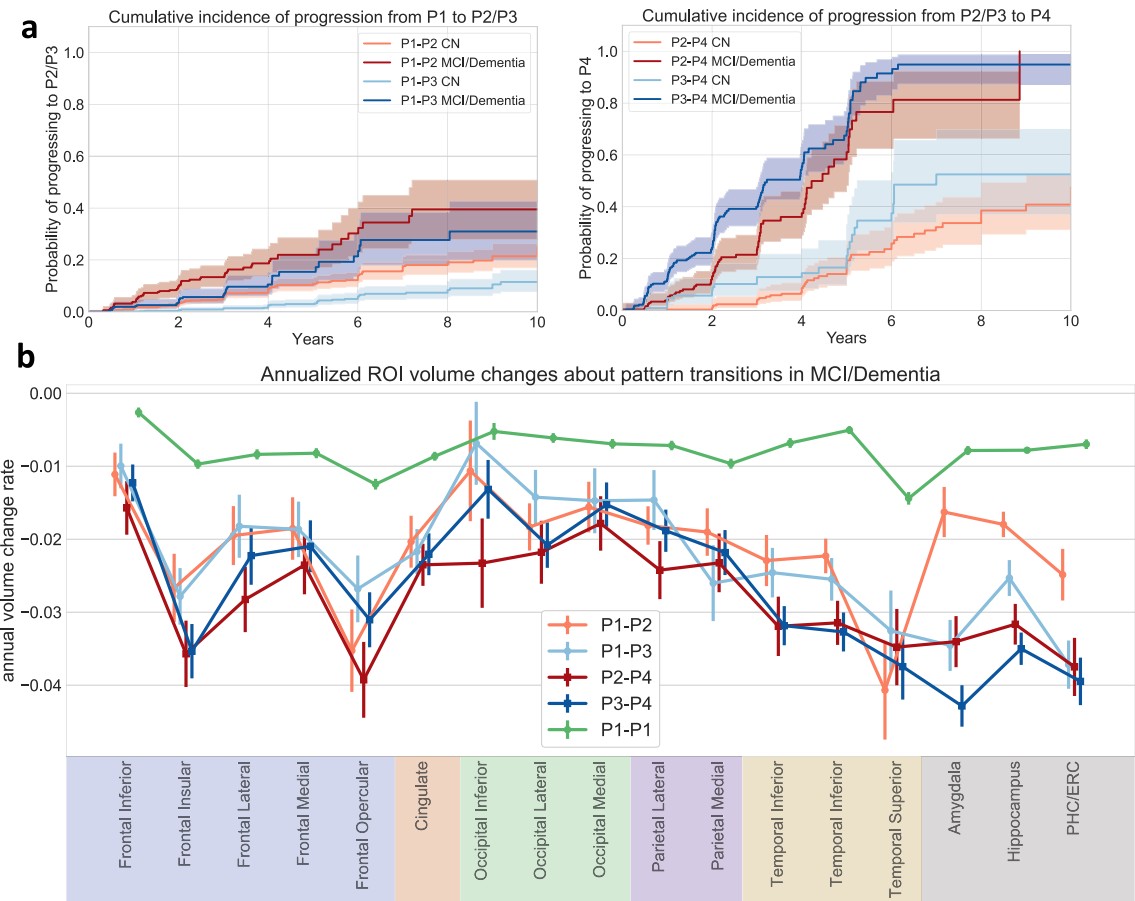

**Fig. 4 Analysis of longitudinal pattern progression.** (Data from all 2832 ADNI/BLSA participants) a Cumulative incidence of pattern progression. The line styles indicate the diagnosis at baseline. 95% confidence intervals are shown with estimated cumulative incidence curves as centres. **b** Annual atrophy rate in selected GM regions along different paths. Data within 3 years before pattern change or last follow-up point (for stable P1 participants (P1-P1)) were utilized and random intercept mixed effect model with time as fixed effect was used to derive annual volume change rate with respect to baseline volume. Data are presented as estimated coefficient of time variable ±standard error. (PHC Parahippocampal gyrus, ERC Entorhinal cortex) Source data are provided as a Source Data file.

those who express P2 at baseline. These relationships hold regardless of cognitive diagnosis at baseline, although baseline CN have much slower rate of pattern progression than those with baseline cognitive impairment. Figure 4b displays differences in volume changes of selected regions among distinct progression pathways. First, participants who persist in P1 show much lower longitudinal atrophy rate in all selected regions. Participants progressing from P1 to P3 show faster medial temporal lobe atrophy while those progressing from P1 to P2 show faster frontal and occipital atrophy. There is an acceleration in medial temporal lobe atrophy associated with the P2–P4 transition. While classified together with a P4 pattern, distinct regional atrophy can be observed between P4 participants who progressed from P2 versus P3, reminiscent of those earlier patterns (Supplementary Fig. 7) and suggesting that the P4 pattern is a common end-stage neurodegeneration pattern.

**Prediction of MRI progression.** Survival curves in Fig. 5a illustrate that participants' baseline pattern expression are associated with the risk of the conversion to P4. Abeta and pTau status at baseline further differentiate higher versus lower risk of future conversion to P4. Moreover, among baseline P1 participants, their baseline P2 and P3 probabilities predict longitudinal progression pathways and progression speed. Using Cox–proportional–hazard models, we found that the baseline probabilities of P2 were able to

discriminate participants with different event time of progressing from P1 to P2 and achieve an average concordance index (C-Index) of $0.823 \pm 0.022$ on the validation set. Similar analyses using baseline P3 probabilities to predict risk of progressing to P3 achieved an average C-index of $0.844 \pm 0.024$. Thus, the baseline P2 and P3 probabilities of P1 participants could imply future risks of progressing to P2 or P3 from 2- to 5-year horizon (see Supplementary Table 9). Prediction performance worsened beyond 5-year risks and the optimal threshold for predicting progression along either pathway decreased with time (see Supplementary Table 9).

**Prediction of clinical progression (change in diagnosis).** Clinical categorizations of CN, MCI, and Dementia provide useful information on functional status. Survival curves in Fig. 5b reveal that, even with similar Abeta or pTau status at baseline, participants with different pattern types show different progression rates for clinical categories. The discrepancy is greater in the MCI to Dementia progression than for the CN to MCI progression, which occurs less frequently across groups. However, only for participants with P2 and P3 at baseline, pTau and Abeta status add significant discrimination power to the risk of converting to Dementia from MCI. Furthermore, pattern probabilities at baseline have comparable predictive power with the SPARE-AD score[10], a previously validated predictive biomarker of AD

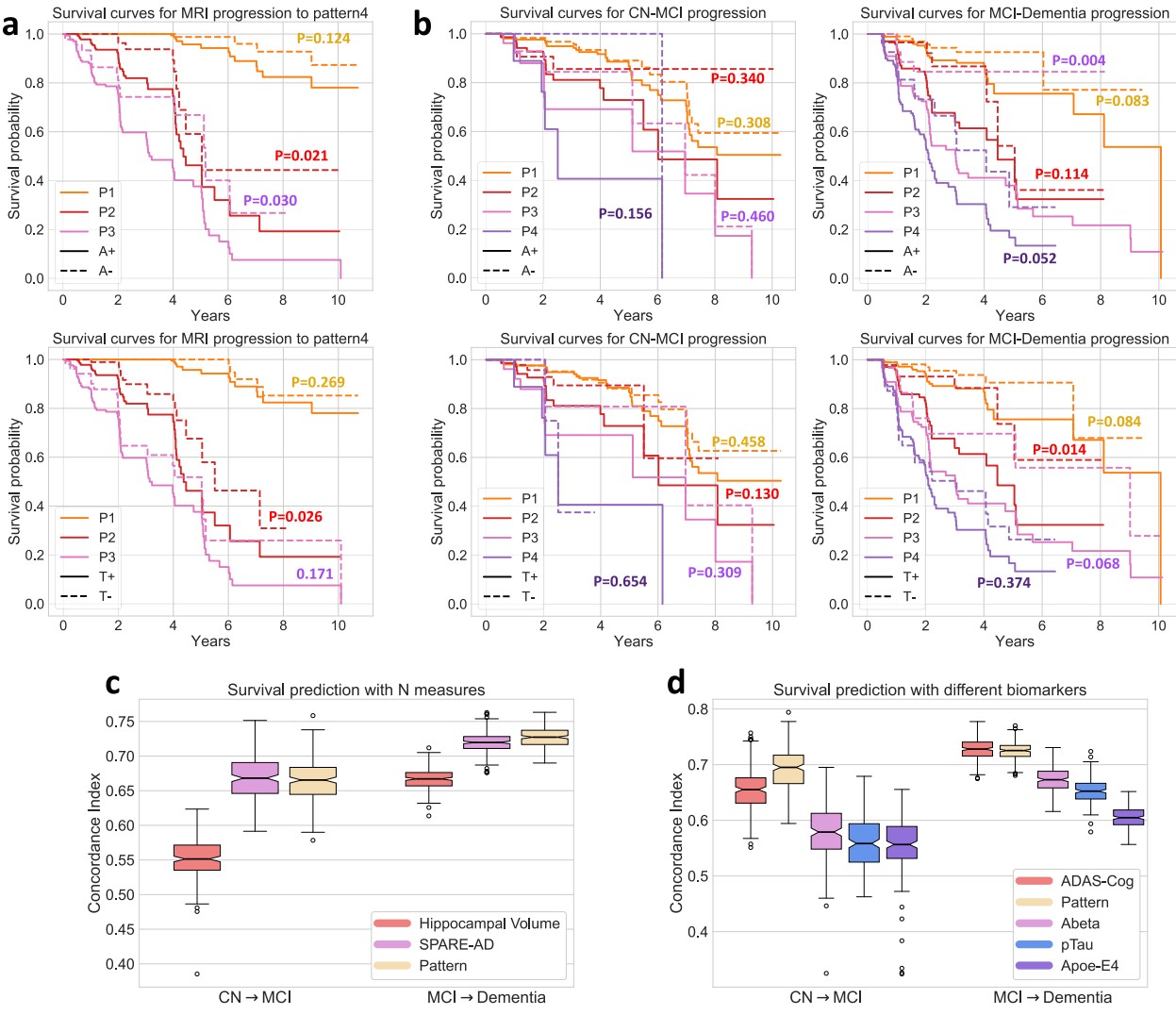

**Fig. 5 Predictive ability of patterns.** (Data from 1194 ADNI participants with Abeta/pTau measures (**a**, **b**, **d**) and 2832 ADNI/BLSA participants (**c**)) **a** Survival curves for neurodegeneration progression to P4; **b** Survival curves for clinical diagnosis progression from CN to MCI and from MCI to Dementia. For both **a** and **b**, survival curves are stratified by both initial dominant pattern and Abeta (A) /pTau (T) status; *p*-values derived from log-rank tests indicate statistical significance of difference between positive and negative Abeta or pTau status within each pattern; **c**, **d** Box and whisker plots of concordance Index (C-Index) which measures the performance of Cox-proportional-hazard model in predicting clinical conversion time (from CN to MCI and MCI to Dementia. Different biomarkers are utilized as features of the model for evaluation of their predictive performance. (Center line, median; box limits, upper and lower quartiles; whiskers, 1.5× interquartile range; points, outliers) Source data are provided as a Source Data file.

neurodegeneration. Both significantly outperformed hippocampal volume (HV, $p < 0.001$ for both SPARE-AD vs HV and Pattern vs HV in CN-MCI and MCI-Dementia prediction, Fig. 5c). Also, compared with other biomarkers including APOE genotype, ADAS-cog score, Abeta and pTau measures, pattern probabilities show either comparable or significantly superior performance in prediction of both CN to MCI and MCI to Dementia progression (Fig. 5d, $p = 0.15$ for pattern vs ADAS-Cog in MCI-Dementia prediction and $p < 0.001$ for comparison between pattern and all other biomarkers).

**Composite score for risk of clinical progression**. With ADAS-Cog score, the most easily ascertained measure, as the only feature, the Cox-proportional-hazard model was able to achieve an average cross-validated C-Index of $0.654 \pm 0.034$ for prediction of CN to MCI progression and $0.728 \pm 0.020$ for prediction of MCI to Dementia progression. Further addition of pattern probabilities derived from T1 MRI significantly boosted average C-Indices for

both tasks to $0.702 \pm 0.042$ ($p < 0.001$) and $0.768 \pm 0.017$ ($p < 0.001$) respectively. Patterns alone provided equivalent or better predictive performance compared to ADAS-cog alone. However, inclusion of Abeta/pTau status, which are derived from either invasive CSF sampling or relatively expensive PET scans, did not bring significant additional improvement to prediction performance (Fig. 6a). With all these biomarkers utilized together, we could construct a composite score indicating risk of clinical progression that was able to predict survival time from MCI to Dementia with an average C-index of $0.785 \pm 0.016$, on randomly split validation sets. Examples of survival curves stratified by the composite score for one randomly split validation set are shown in Fig. 6b.

**Discussion**
We have developed a deep-learning approach, Smile-GAN, which disentangles pathologic neuroanatomical heterogeneity and defines subtypes of neurodegeneration by learning to generate

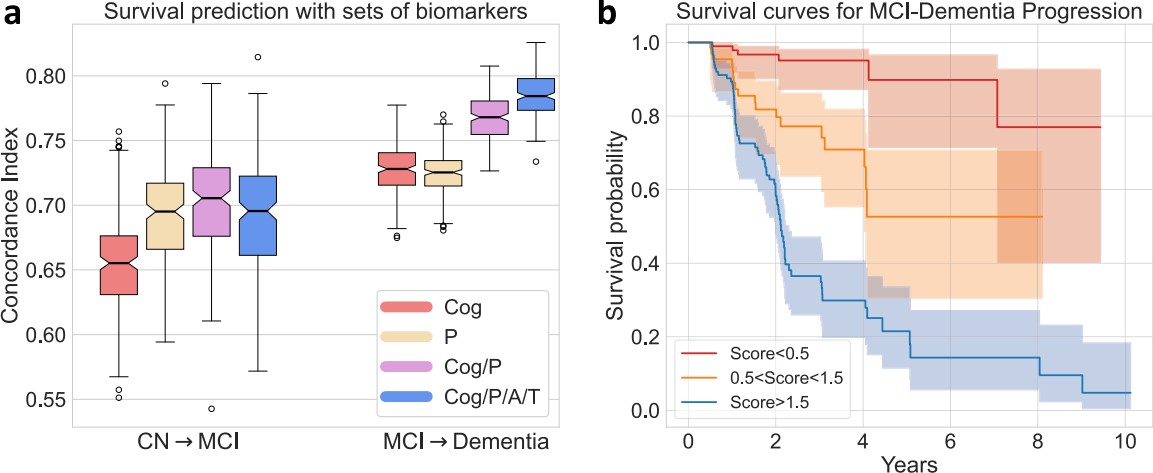

**Fig. 6 Prediction of clinical diagnosis progression with composite biomarkers.** Data from 1194 ADNI participants with Abeta/pTau measures. **a** Biomarkers were added successively into features set based an order of accessibility. Concordance Index (CI) measures the performance of Cox-proportional-hazard model in predicting clinical conversion time (from CN to MCI and MCI to Dementia) Different sets of biomarkers are utilized as features of the model for evaluation of their predictive powers. (Center line, median; box limits, upper and lower quartiles; whiskers, 1.5× interquartile range; points, outliers). **b** Survival curves stratified by composite scores (A, T, Pattern, ADAS-Cog jointly predicting outcome in cross-validated fashion) for one randomly split validation set. 95% confidence intervals are shown with estimated survival curves as centres. (A: Abeta; T: pTau, P: Pattern, Cog: ADAS-Cog score) Source data are provided as a Source Data file.

mappings from regional volume data of cognitively normal individuals to that of patients. Compared with unsupervised methods[12–14], Smile-GAN has an advantage in avoiding non-disease-related confounding variations, thereby identifying neuroanatomical patterns associated with pathology. This stems from the fundamental property of Smile-GAN to cluster the transformations from normal to pathologic anatomy, rather that clustering patient data directly. Also, the deep-learning-basedSmile-GAN can easily handle high dimensional ROI data. Thus, no preprocessing ROI selection is required, and the model is able to fully capture variations in all subdivided ROIs and could feasibly be extended to even smaller/more numerous ROI. Moreover, in contrast with other semi-supervised methods[17,18], Smile-GAN makes no assumption about data distribution and data transformation linearity, and in validation experiments was found to be robust to mild, sparse, or overlapping patterns of pathology (neurodegeneration, herein). Critically, pattern probabilities given by Smile-GAN are easily interpretable continuous biomarkers reflecting the neuroanatomical expression of respective patterns. These advantages of Smile-GAN allow versatile characterization of pattern types related to both severity and heterogeneity of pathological effects.

Smile-GAN does not intrinsically model disease stage, nor does it assume any progression pathway between patterns. Staging can be inferred at a second-level analysis of the degree of expression of each of the identified patterns, or a linear or nonlinear combination of them. In ADNI/BLSA, most individuals expressed multiple patterns at the same time, as reflected by the magnitude of various pattern probabilities, an observation similar to the findings of Zhang et al.[15]. This feature allows investigation of complex and nonlinear relationships between pattern-based stage and clinical outcomes of interest, which can vary depending on the outcome (e.g., from various cognitive or clinical measures to staging estimates that inform clinical trial recruitment).

Application of Smile-GAN to MRI data from a sample enriched with AD pathology identified 4 patterns of regional brain atrophy expressed in participants across the AD spectrum, which were highly reproducible on validation experiments, including a permutation test. These patterns range from mild to advanced atrophy and define two progression pathways. One pathway, here

termed the P1-3-4 pathway, shows early atrophy in the medial temporal lobe that is typical for AD. The second pathway, P1-2-4, shows early diffuse mild cortical atrophy with MTL sparing that is a less typical pattern for AD. The end stage for both pathways is an advanced atrophy pattern, P4. This four-pattern system has similarities with other neuroimaging-based clustering studies, including identification of temporal and cortical predominant patterns[12,15,25,26]. Smile-GAN patterns tentatively correspond to pathologically identified subtypes of AD[24]: Limbic Predominant, matching P3, Hippocampal Sparing, matching P2, and typical AD (mixed P2–P3). There is one another possible subtype of subcortical atrophy previously identified by several other MRI-based unsupervised clustering algorithms using ADNI data[12,15] but not distinctly identified by Smile-GAN. There are a few reasons why Smile-GAN did not identify a subcortical pattern. First, as observed in Zhang et al.[27], the subcortical pattern may merge with the temporal pattern based upon harmonization and clustering methodology and specific training sample. Second, a portion of the variability attributed to a subcortical atrophy may not be disease-related, resulting in insufficient signal to separately cluster as a distinct pattern, a possibility potentially supported by the lack of pathological evidence for this subtype and scarce atrophy in subcortical ROIs among the patient group (Supplementary Fig. 6).

The four Smile-GAN patterns have clinically meaningful implications. Pattern membership is associated with differences in cognitive test performance, with P2 having relatively more executive dysfunction, P3 showing greater memory impairment, and P4 showing the worst performance across domains. These patterns also have implications for speed and direction of progression, with early pattern features predictive of the future pattern of neurodegeneration and pattern features predictive of clinical progression from CN to MCI and MCI to dementia. Critically, pattern expression was the most important predictor of clinical progression, showing comparable or stronger predictive ability than other N measures and biomarkers (Fig. 5). Synergistically, pattern expression, A, T and ADAS-Cog provided outstanding cross-validated prediction of clinical progression on an individual basis (Fig. 6b), underlining the potential significance of this combined predictive index for patient

management, for clinical trial recruitment, and for evaluation of treatment response.

While the patterns are relatively distinct in regional specificity and severity, the underlying pathophysiology is more complex. P1 indicates that no significant neurodegeneration is present. Yet a significant number of participants ($N = 306$) with dominant P1 pattern still had objective cognitive impairment with MCI, and even a few cases of dementia, both with and without evidence of amyloid and tau deposition. These participants likely have reduced cognitive reserve and/or non-neurodegenerative contributions to MCI/dementia. The P2 group shows mild diffuse atrophy and is likely a group inclusive of multiple mild or early pathologies, inclusive of hippocampal sparing/cortical presentations of AD and other early neurodegenerative processes or atrophy related to chronic systemic disease, in part evidenced by amyloid negative P2 participants. However, P2 is not disproportionately enriched for vascular disease, a common comorbidity for primary neurodegenerative diseases, at least as measured by WML volumes which were relatively similar across P2–P3–P4. Regardless of etiology, expression of a P2 pattern is akin to concepts of advanced brain aging or decreased brain reserve[5]. While P3 is predominately early typical AD within the enriched ADNI sample, this also likely includes other pathologies such as limbic-predominant age-related TDP-43 encephalopathy (LATE)[28]. P4 appears to be a composite of advanced or 'end-stage' neurodegeneration patterns. While fully typical of advanced AD, this pattern is also seen in participants with cognitive decline without amyloid or tau deposition, indicating a late-stage similarity of widespread brain atrophy across multiple pathologies.

With the growing utilization of the AT(N) framework[29], these patterns provide a means to quantify neurodegeneration into a few informative categories rather than as a binary measure. Categorization of neurodegeneration as absent (P1), early cortical (P2), temporal-predominant (P3) or advanced (P4) provides important phenotypic information while preserving much of the simplicity of the binary AT(N) framework. Together with A/T status, the dynamics of pattern expression shows both severity of disease and identifies reasonably distinct and reasonably sized subgroups with differing balance of AD and non-AD pathology (Fig. 7). These groups could be used to enrich for typical AD pathology for clinical trials, reduce the need for ascertaining certain biomarkers, and identify interesting subgroups for focused evaluation, such as for genetic factors of resilience. For example, to recruit a group with early, typical AD neurodegeneration, one could initially select those with P3 pattern on MRI (a group that is 25.3% A+T+ in this study sample) and ascertain A/T biomarkers only in this group.

The Smile-GAN pattern approach has several advantages. It captures biologically relevant regional atrophy patterns that are few in number, providing meaningful, top-level detail on neurodegeneration without requiring significant complexity, while simultaneously maintaining quantitative pattern probability information. The Smile-GAN method is a data-driven approach that can be applied on features extracted from data beyond neuroimaging, potentially able to cluster patients effectively based on any selected disease-related feature changes from normal group to patient group. Therefore, it is generalizable to any diseases and disorders that have reproducible patterns of changes in imaging or other biomedical data, including but not limited to other neurodegenerative and neuropsychiatric diseases[30]. While there are modest time and computational requirements for training each model primarily dependent on the number of features, the training process is only performed once, and subsequent calculation of individual pattern scores using an existing model is rapid. There are limitations to the method and our implementation. First, selection of the control group exerts a

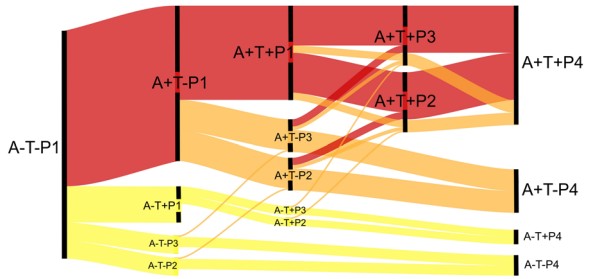

**Fig. 7 Hypothetical flow diagram of implications of pattern pathways on the ATN framework.** Cascade of biomarkers can follow a canonical AD pathway, which is the most represented in the ADNI sample (red). The relationships of patterns with amyloid/tau status identifies another large group with the presence of AD pathology and significant or even dominant copathology (orange) as well as groups with suspected non-AD pathology (yellow). These pathways also indicate that certain typical AD neurodegenerative phenotypes may in some cases be driven by copathology. For example, A+T+ nodes are typical for AD; however, there are several potential paths (orange) whereby copathology may be the dominant cause of the neurodegenerative pattern. Path thickness estimates approximate flux through nodes in ADNI. This model is based on distribution of cross-sectional data in A/T/P categories and the assumption that events happen in certain order (A-→A+; T-→T+; P1→P2→P4 and P1→P3→P4).

critical influence on resultant patterns, since, by design, any changes that are common in the control group will not be distinctly segregated (Supplementary Fig. 1). For example, this may be a reason that vascular disease was distributed across multiple patterns. Similarly, rare and/or subtle patterns of atrophy may not be distinctly learned by the model, as demonstrated in simulation experiments. It is possible that larger and more diverse training data may allow identification of more pattern types in the AD continuum. Thresholds for assigning participants to groups may benefit from optimizations tailored to specific hypotheses. The performance of the four-pattern model in this study was derived and evaluated using data from the ADNI and BLSA studies, which have high and low prevalence of AD, respectively, and relatively low prevalence of non-AD neurodegeneration. Direct application of this model to a memory-center population with mixed neurodegenerative disease has not been evaluated. Finally, the Smile-GAN model is currently applied to ROI volume data derived from MRI images only, and thus may fail to capture more subtle patterns that do not conform to anatomic ROIs. The Smile-GAN model architecture is flexible for use with smaller ROI parcellations or voxel-based analyses as well as non-structural MRI and non-imaging data. Extension of current framework to such other types of data is a direction for future development.

Patterns identified using semi-supervised clustering with generalized adversarial networks provide useful information about the severity and distribution of neurodegeneration across the AD spectrum. Baseline patterns are predictive of the future pattern of neurodegeneration as well as clinical progression to MCI and dementia. These patterns could augment research and clinical assessments of participants and patients with cognitive decline and contribute to a dimensional characterization of brain diseases and disorders.

## Methods

**Smile-GAN model.** Smile-GAN is a Generative Adversarial Network (GAN) architecture for clustering a group (in our case patients) based on their multivariate differences (in our case regional volumes derived from MRI) to a reference group (in our case healthy controls). The general structure of Smile-GAN is shown

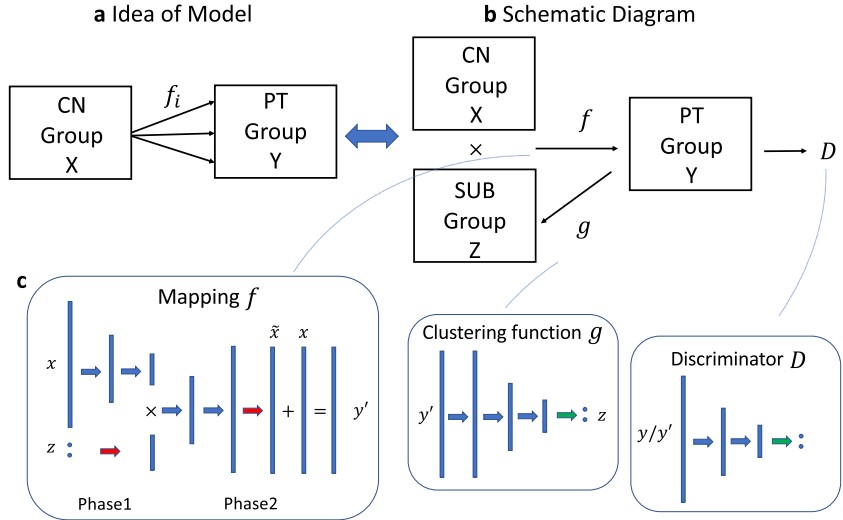

**Fig. 8 Schematic diagram and network architectures. a** General idea behind Smile-GAN. The model aims to learn several mappings from the CN group to the PT group **b** Schematic diagram of Smile-GAN. The idea of the model is realized by learning one mapping from joint of two groups $X \times Z$ to $Y$, while learning another function $g : Y \rightarrow Z$. CN cognitive normal control, PT patient, Sub pattern subtype. **c** Network architecture of three functions: blue arrow represents one linear transformation followed by one leaky rectified linear unit function, green arrow represents one linear transformation followed by one softmax function, red arrow represents only one linear transformation.

in Fig. 8. To sum up, the primary concept of the model is to learn one-to-many mappings from the CN group $X$ (alternatively called domain $X$: set of CN data) to the patient (PT) group $Y$ (alternatively called domain $Y$: set of PT data). The idea is equivalent to learning one mapping function, f : $X \times Z \rightarrow Y$, which generates synthesized PT data $\mathbf{y'} = f(\mathbf{x}, \mathbf{z})$ from the real CN data and sampled subtype variable $z$, while enforcing the indistinguishability between PT data and synthesized PT data. Put simply, given one same value for subtype variable,$\mathbf{z}$, the mapping $f(\cdot)$ generates image data that match data of patients of similar subtype mix. Here, $Z = \{\boldsymbol{\alpha} : \sum_{i=1}^{M} \boldsymbol{\alpha}_i = 1\}$, referred as the subtype (SUB) group (alternatively called SUB domain), is a class of vectors with dimension $M$ ($M = 4$ was found to be optimal in our experiments). We denote the distribution of the aforementioned variables as $\mathbf{x} \sim p_{CN}$, $\mathbf{y} \sim p_{PT}$, $\mathbf{y'} \sim p_f$, $\mathbf{z} \sim p_{Sub}$, respectively. The variable $\mathbf{z}$, independent from $\mathbf{x}$, takes values from a subclass of group $Z$ and can be encoded as a one-hot vector with value 1 being placed at any position with equal probability (i.e., $1/M$). In addition to the mapping function, an adversarial discriminator $D$ is introduced to distinguish between real PT data $\mathbf{y}$ and synthesized PT data $\mathbf{y'}$, thereby ensuring that the mappings f generate image data that are indistinguishable from real patient data.

The fact that a number of functions can potentially achieve equality in distributions makes it hard to guarantee that the mappings learned by the model are closely related to the underlying pathology progression. Moreover, during the training procedure, the mapping function backboned by the neural network tends to trivially ignore the Sub variable $\mathbf{z}$. Therefore, with the assumption that there is one true underlying function for real PT variable $\mathbf{y} = h(\mathbf{x}, \mathbf{z})$, Smile-GAN aims to boost the mapping function f to be approximate to the true underlying function h, by constraining the function class via three types of regularization: (1) we encourage sparse transformations, (2) enforce Lipschitz continuity of functions, (3) introduce another function g : $Y \rightarrow Z$ to the model structure. The latter is a critical part of the algorithm's ability to cluster the data, as it requires that the mapping functions identify sufficiently distinct imaging patterns in the $Y$ group, which would allow the inverse mapping $g(\cdot)$ to estimate the correct subtype in the PT group. More details about regularization terms and clustering inference of function g are stated in Supplementary Method 1.

The objective of Smile-GAN is a combination of adversarial loss[19] and regularization terms. First, the adversarial loss[19] aims at matching the distribution synthesized PT data, $p_f$, to the distribution of real PT data, $p_{PT}$, which can be denoted as:

$$L_{GAN}(D, f) = E_{\mathbf{y} \sim p_{PT}}[\log(D(\mathbf{y}))] + E_{\mathbf{z} \sim p_{Sub}, \mathbf{x} \sim p_{CN}}[1 - \log(D(f(\mathbf{x}, \mathbf{z})))] \quad (1)$$

$$= E_{\mathbf{y} \sim p_{PT}}[\log(D(\mathbf{y}))] + E_{\mathbf{y'} \sim p_f}[1 - \log(D(\mathbf{y'}))] \quad (2)$$

where the mapping f attempts to transform CN to synthetically generated PT data so that they follow similar distributions as real PT data. The discriminator D, providing a probability that $\mathbf{y}$ comes from the real data rather than the generator, is trying to identify the synthesized PT data and distinguish it from the real PT data. Therefore, the discriminator attempts to maximize the adversarial loss function while the mapping f attempts to minimize against it. The corresponding training process can be denoted as:

$$\min_f \max_D L_{GAN}(D, f) = E_{\mathbf{y} \sim p_{PT}}[\log(D(\mathbf{y}))] + E_{\mathbf{y'} \sim p_f}[1 - \log(D(\mathbf{y'}))] \quad (3)$$

Second, the regularization terms include the change loss and cluster loss, both serving to constrain the function space where f is learned from. The change loss is defined as:

$$L_{change}(f) = E_{\mathbf{x} \sim p_{CN}, \mathbf{z} \sim p_{Sub}}[\|f(\mathbf{x}, \mathbf{z}) - \mathbf{x}_1\|] \quad (4)$$

By denoting $l_c$ to be the cross-entropy loss with $l_c(\mathbf{a}, \mathbf{b}) = -\sum_{i=1}^{k} \mathbf{a}^i \log \mathbf{b}^i$, we define the cluster loss as:

$$L_{cluster}(f, g) = E_{\mathbf{x} \sim p_{CN}, \mathbf{z} \sim p_{Sub}}[l_c(\mathbf{z}, g(f(\mathbf{x}, \mathbf{z})))] \quad (5)$$

With the aforementioned losses, we can write the full objective as:

$$L(D, f, g) = L_{GAN}(D, f) + \mu L_{change}(f) + \lambda L_{cluster}(f, g) \quad (6)$$

where $\mu$ and $\lambda$ are two hyperparameters that control the relative importance of each loss function during the training process. Through this objective, we aim to find the mapping function f and clustering function g such that:

$$f, g = \arg \min_{f,g} \max_D L(D, f, g) \quad (7)$$

More implementation details of the model, including network architecture, training details, algorithm, and training stopping criteria are presented in Supplementary Method 2.

**Study and participants**. The Alzheimer's Disease Neuroimaging Initiative (ADNI, http://www.adni-info.org/) study is a public-private collaborative longitudinal cohort study which has recruited participants categorized as cognitively normal, MCI, and AD participants through 4 phases (ADNI1, ADNIGO, ADNI2)[31]. ADNI has acquired longitudinal MRI, cerebrospinal fluid (CSF) biomarkers, and cognitive testing. The Baltimore Longitudinal Study of Aging, neuroimaging substudy, has been following participants who are cognitively normal at enrollment with imaging and cognitive exams since 1993. A total number of 1718 ADNI participants (819 ADNI1 and 899 ADNIGO/ADNI2) and 1114 BLSA participants were included in the study. Detailed information of enrollment criteria can be found in Peterson et al.[32] for ADNI and Resnick et al.[4] for BLSA. Details of both studies including number classified as CN/MCI/Dementia at baseline, number of participants with CSF Abeta/Tau biomarkers, length of follow-up, age, gender, APOE genotype are included in Table 1. Participants provided written informed consent to the ADNI and BLSA studies. The protocol of this study was approved by the University of Pennsylvania institutional review board.

**MRI data acquisition and processing**. 1.5 T and 3T MRI data were acquired from both ADNI and BLSA study introduced above. A fully automated pipeline was applied for processing T1 structural MRIs. T1-weighted scan of each participant is first corrected for intensity inhomogeneities[33]. A multi-atlas skull stripping algorithm was applied for the removal of extra-cranial material[34]. For the ADNI study, 145 anatomical regions of interest (ROIs) were identified in gray matter (GM, 119

**Table 1 Details of ADNI and BLSA studies.**

| Study | CN | MCI | Dementia | Median follow-up (years) | Gender (% of male) | Age | APOE E4 carriers | CSF Abeta/Tau available |
|-------|-----|-----|----------|--------------------------|--------------------|-----|------------------|-------------------------|
| ADNI1 | 229 | 397 | 193 | 2.2 (1.7–3.1) | 58.2% | 75 (71–80) | 48.8% | 415 |
| ADNI2/ GO | 297 | 452 | 150 | 2.1 (1.1–4.0) | 53.3% | 74 (68–79) | 43.6% | 779 |
| BLSA | 1094 | 11 | 9 | 4.0 (0.0–6.0) | 47% | 67 (58–76) | 25% | 0 |

For age and length of follow-ups, median value with first and third quartile are reported. APOE E4 carriers include heterozygotes and homozygotes.

ROIs), white matter (WM, 20 ROIs) and ventricles (6 ROIs) using a multi-atlas label fusion method[35]. For the BLSA study, this method was combined with harmonized acquisition-specific atlases[36] to derive the same 145 ROIs. Phase-level cross-sectional harmonization was applied on regional volumes of the 145 ROIs to remove site effects[37]. For visualization of disease patterns, tissue density maps, referred as RAVENS (regional analysis of volumes examined in normalized space[38]) were computed as follows. Individual images were first registered to a single subject brain template and segmented into GM and WM tissues. RAVENS maps encode, locally and separately for each tissue type, the volumetric changes observed during the registration.

**Data separation and preparation.** After preprocessing, baseline ROI data of 297 CN and 602 cognitively impaired participants from ADNI2/GO participants were selected as the discovery set for training and validation of the model. longitudinal ROI data from follow-up visits of all participants from ADNI and BLSA were used for further clinical analysis, including both participants whose baseline data were used for model training and those who were completely independent of the discovery set. For analysis requiring measures of CSF Abeta/pTau, only ADNI participants with these two biomarkers were included. Otherwise, all participants from ADNI and BLSA study were incorporated for analysis.

Before being used as features for the Smile-GAN model, ROI volumes were residualized and variance-normalized. To correct age and sex effects while keeping disease-associated neuroanatomical variations, we estimated ROIs-specific age and sex associations among 297 CN participants using a linear regression model. All cross-sectional and longitudinal data were then residualized by age and sex effects. Then, all ROI volumes were further normalized with respect to 297 CN participants in the discovery set to ensure a mean of 1 and standard deviation of 0.1 among CN participants for each ROI.

**Cognitive, clinical, CSF biomarker, and genetic data.** We used additional clinical, biofluid, and genetic variables, including CSF biomarkers of amyloid and tau, APOE genotype, and cognitive test scores, provided by ADNI. These measures were downloaded from the LONI website. A total of 1194 participants from ADNI have CSF measurements of ß-amyloid, total tau, and phospho-tau, including 383 CN, 578 MCI and 233 Dementia at baseline; BLSA participants do not have A/T biomarkers and are therefore excluded from analyses based upon those measures. Detailed methods for CSF quantification are described in Hansson et al.[39]. Cutoffs for amyloid status based on ß-amyloid measures and for tau status based upon phospho-tau measures were previously defined[39] and used to categorize participants as positive or negative for cerebral amyloid and tau deposition. Tau measures are also presented as continuous variables. Composite cognitive scores across several domains have been previously validated in the ADNI cohort. The memory composite (ADNI-MEM) models based on components from the Rey Auditory Verbal Learning Test, Alzheimer's Disease Assessment Scale–Cognitive Subscale (ADAS-Cog), and mini-mental status exam (MMSE)[40]. The executive function composite (ADNI-EF) models based on animal and vegetable category fluency, trail-making A and B, digit span backwards, digit symbol substitution from the revised Wechsler Adult Intelligence Scale, and circle, symbol, numbers, hands, and time items from a clock drawing task[41]. The language composite (ADNI-LAN) models using animal and vegetable category fluency, the Boston naming total, MMSE language elements, following commands/object naming/ideational practice from ADAS-Cog, and Montreal Cognitive Assessment (MoCA) language elements, including letter fluency, naming, and repeating tasks[42]. Further detail on these composite measures can be obtained on the ADNI website (https://adni.bitbucket.io/reference/docs/UWNPSYCHSUM/adni_uwnpsychsum_doc_20200326.pdf).

White matter lesion (WML) volumes were calculated from both ADNI and BLSA using inhomogeneity-corrected and co-registered FLAIR and T1-weighted images and a deep-learning-based segmentation method[43] built upon the U-Net architecture[44], with the convolutional layers in the network replaced by an Inception ResNet architecture[45]. The model was trained using a separate training set with human-validated segmentation of WML. WML volumes were first cubic rooted. Then phase-level cross-sectional harmonization was applied on them to reduce site effects.

**Pattern memberships and probabilities assignments.** Smile-GAN model assigns $M$ probability values to each participant, with each probability corresponding to one pattern type and the sum of $M$ probabilities being 1. Based on the $M$

probability values, we can further assign each participant to the dominant pattern type, determined by the maximum probability. The optimal $M$ was chosen during a cross-validation (CV) procedure based on the clustering reproducibility or stability. Specifically, we ran 10 folds of repeated holdout CV for $M = 3$ to 5. For each fold, we randomly left out 20% of the discovery set to add variability. Of note, $M = 2$ generally stratified the data into mild and severe atrophy patterns, which is not clinically interesting. We used the Adjusted Rand Index (ARI)[46] to quantify the clustering stability of the 10 folds/models. ARI is a corrected for chance version of the random index which equals 0 for two random partitions and is, thus, considered a good choice for measuring overlap of clustering results in our case. The highest mean pair-wise ARI, $0.48 \pm 0.08$, was reached at $M = 4$, with ARI $= 0.30 \pm 0.12$ for $M = 3$ and ARI $= 0.33 \pm 0.07$ for $M = 5$. A permutation experiment demonstrated significant reproducibility of the Smile-GAN patterns for $M = 3$–5, as measured by ARI. Together, these data suggested that $M = 4$ yields the optimal number of clusters (Supplementary Section 1.4/2.4).

With $M = 4$, we reran Smile-GAN 30 times with all available data in the discovery set and the trained models will be used for external validation and analysis. In order to find the best correspondence among cluster assignments across the 30 experiments, we calculated the mean pair-wise ARI values for each resultant model. The one with the highest ARI was chosen as the template and the pattern types learned by all other models were reordered so that their clustering results achieved the highest overlap with that of the template. After reordering, the average probability of each pattern across all 30 models was taken as the probability of the corresponding pattern for each participant. We then applied these learned models to longitudinal data of all CN/MCI/Dementia participants and obtained probabilities of four patterns for all visits of each participant.

**Statistical analysis.** To visualize the brain signatures of four patterns, we utilized all cross-sectional data of MCI/Dementia participants in the discovery set and performed voxel-wise group comparisons (i.e., CN vs each pattern) via AFNI 3dttest[47] using voxel-wise tissue density (RAVENs) maps[38]. To access longitudinal progression trajectories of pattern assignment, we grouped for each of the four patterns those participants with probability larger than 0.5. We then compared how the pattern probabilities change over time for each of the four groups by calculating pattern probability for P1–P4 of all within group who have data available in a given time interval (i.e., X year–X + 1 year). Those who had more than one data point in the selected time interval only contributed once through mean probabilities of all those visits. The demographic variables, APOE genotype, CSF biomarker levels, cognitive test scores, WML volumes and pattern probabilities were compared both across pattern types and within pattern types. Only participants from the ADNI study whose Abeta/pTau status was available at baseline were included for comparison. For categorical variables, the Chi-squared test was used to identify differences between subgroups. For other quantitative variables, a one-way ANOVA analysis was performed for group comparison. Statistical analyses were conducted via online python packages, statsmodels 0.8.0, SciPy 1.6.3, NumPy 1.16.6 and pandas 0.21.0.

To assess the risk of converting from P1 into P2 or P3, we conducted time-to-event survival analysis to evaluate the risk pattern conversions. In particular, we treated P2 and P3 as competing events and used Aalen-Johansen estimator to generate cumulative incidence curves for P1 to P2/P3 progression. For all other cumulative incidence curves and survival curves corresponding to pattern progression and diagnosis transformation, we applied a nonparametric Kaplan–Meier estimator and used the log-rank test to compare difference in survival distributions between groups.

For all survival analysis, participants were assigned into one pattern at baseline or labeled as progressing to one pattern only if the corresponding pattern probability is greater than 0.5. A few participants not reaching this threshold in any pattern at baseline were discarded to avoid noise in the analysis. All survival analyses were conducted via online python package lifelines 0.25.7.

**Evaluation of patterns' predictive ability.** We further conducted analyses to evaluate the predictive ability of baseline pattern probabilities in the prediction of future pattern changes. Also, we compared them with other measures of neurodegeneration (N measures) and clinical biomarkers in prediction of diagnosis transitions.

For pattern progression prediction, we selected all 940 participants who had longitudinal follow-ups and P1 > 0.7 at baseline to avoid trivial prediction tasks. First, the Cox-proportional-hazard model with baseline P2 or P3 probability as the

only feature was utilized to predict survival curves from P1 to P2 or P1 to P3, respectively. We ran the two-fold cross validation 100 times and derived the concordance index on validation sets. Second, to predict risk of pattern progression and progression pathways of P1 participants at specific time points X, we directly used P2 probability and P3 probability at baseline as an indication of risk without further fitting any additional models. For each time X from 2 years to 8 years, we generated a binary indicator with 0 representing not progressing to P2 till T and 1 representing who have already progressed to P2 before T and directly used baseline P2 probabilities to discriminate these two groups. The exact same process was also done for P3. Area under the receiver operator characteristic curve (AUC) values were calculated for both P2 and P3 at different time X. Optimal discrimination thresholds, at which true positive rate (TP) plus false positive rate (FP) = 1, were reported for two different progression pathways.

To predict clinical diagnosis changes, we selected out 1178 CN participants and 921 participants categorized as MCI at baseline who had longitudinal follow-ups. First, to compare Patterns with other N measures, we again utilized the Cox-proportional-hazard model with different N measures as features to predict CN-MCI and MCI-Dementia survival curves. Two-fold cross validation was run 100 times to derive the concordance index on validation sets. Then, to compare the prognostic powers of Pattern, Abeta, pTau, APOE genotype and ADAS-Cog, we reduced samples to 380 CN and 568 MCI participants who had these biomarkers. Each biomarker was used independently as the only feature for training the model.

For all prediction tasks in this section, baseline pattern assignments and progression labelling followed the same rule introduced in the 'Statistical Analysis' section if not specifically annotated.

**Biomarker selection and composite score construction**. Finally, we evaluated predictive powers of different combinations of biomarkers mentioned above. Following the order of accessibility, ADAS-Cog, pattern probabilities derived from T1 MRI, Abeta/pTau derived from PET scan were added successively to the feature set for training the Cox-proportional-hazard model and the same experimental procedure were implemented as introduced in the previous section. A composite score indicating the risk of clinical progression can be derived with all biomarkers introduced above. Using Pattern-probabilities/Abeta/pTau/ADAS scores at baseline as features, the trained Cox-proportional-hazard model was applied to the validation set to derive the partial hazard as the composite score.

**Reporting summary**. Further information on research design is available in the Nature Research Reporting Summary linked to this article.

## Data availability

Data used for this study were provided from ADNI and BLSA studies via data sharing agreements that did not include permission to further share the data. Data from ADNI are available from the ADNI database (adni.loni.usc.edu) upon registration and compliance with the data usage agreement. Data from the BLSA are available upon request from the BLSA website (blsa.nih.gov). All requests are reviewed by the BLSA Data Sharing Proposal Review Committee and may also be subject to approval from the NIH Institutional Review Board. Those interested in accessing study data or derived imaging variables used in this study may seek approval from studies. If granted, we would be able to provide participant-level derived imaging variables used in this study within 1 month of approval. Source data are provided with this paper.

## Code availability

The software Smile-GAN is available as a published PyPI package. Detailed information about software installation, usage, and license can be found at: https://pypi.org/project/SmileGAN/. Custom code can be found at: https://github.com/zhijian-yang/SmileGAN.

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

## Acknowledgements

The iSTAGING consortium is a multi-institutional effort funded by NIA by RF1 AG054409. The Baltimore Longitudinal Study of Aging neuroimaging study is funded by the Intramural Research Program, National Institute on Aging, National Institutes of Health and by HHSN271201600059C. This study was also supported in part by grants from the National Institutes of Health (U19-AG033655). Data used in preparation of this article were in part obtained from the Alzheimer's Disease Neuroimaging Initiative (ADNI) database (adni.loni.usc.edu). As such, the investigators within the ADNI contributed to the design and implementation of ADNI and/or provided data but did not participate in analysis or writing of this report. A complete listing of ADNI investigators can be found at: http://adni.loni.usc.edu/wpcontent/uploads/how_to_apply/ADNI_Acknowledgement_List.pdf. ADNI is funded by the National Institute on Aging, the National Institute of Biomedical Imaging and Bioengineering, and through generous contributions from the following: AbbVie, Alzheimer's Association; Alzheimer's Drug Discovery Foundation; Araclon Biotech; BioClinica, Inc.; Biogen; Bristol-Myers Squibb Company; CereSpir, Inc.; Cogstate; Eisai Inc.; Elan Pharmaceuticals, Inc.; Eli Lilly and Company; EuroImmun; F. Hoffmann-La Roche Ltd and its affiliated company Genentech, Inc.; Fujirebio; GE Healthcare; IXICO Ltd.; Janssen Alzheimer Immunotherapy Research & Development, LLC.; Johnson & Johnson Pharmaceutical Research & Development LLC.; Lumosity; Lundbeck; Merck & Co., Inc.; Meso Scale Diagnostics, LLC.; NeuroRx Research; Neurotrack Technologies; Novartis Pharmaceuticals Corporation; Pfizer Inc.; Piramal Imaging; Servier; Takeda Pharmaceutical Company; and Transition Therapeutics. The Canadian Institutes of Health Research is providing funds to support ADNI clinical sites in Canada. Private sector contributions are facilitated by the Foundation for the National Institutes of Health (www.fnih.org). The grantee organization is the Northern California Institute for Research and Education, and the study is coordinated by the Alzheimer's Therapeutic Research Institute at the University of Southern California. ADNI data are disseminated by the Laboratory for Neuro Imaging at the University of Southern California.

## Author contributions

Study design: Z.Y., I.N. and C.D. Model Development: Z.Y. Statistical analysis: Z.Y. and I.N. Data interpretation: Z.Y., I.N., H.S., J.W., M.H., G.E., A.A., S.R., D.W. and C.D. Data collection and processing: I.N., H.S., J.D., G.E., A.A., S.R., M.A., P.M., J.F., J.M., D.W. and C.D. Figure generation: Z.Y. and I.N. Literature search: Z.Y., I.N. and J.W. Manuscript writing: Z.Y., I.N. and C.D. Manuscript critical revision and submission approval: Z.Y., I.N., H.S., J.W., J.D., M.H., G.E., A.A., S.R., M.A., P.M., J.F., J.M., D.W. and C.D.

## Competing interests

We disclose here that I.N. served as an educational speaker for Biogen. The remaining authors declare no competing interests

## Additional information

## iSTAGING Consortium

Christos Davatzikos[1,3✉], David A. Wolk[1,11,12], Yong Fan[1,3], Haochang Shou[1,4], Ahmed Abdulkadir[1,3], Guray Erus[1,3], Vishnu Bashyam[1,3], Jimit Doshi[1,3], Mohamad Habes[1,5], Elizabeth Mamouiran[1,3], Randa Melhem[1,3], Ilya M. Nasrallah[1,3,100], Raymond Pomponio[1,3], Dushyant Sahoo[1,3], Singh Ashish[1,3], Ioanna Skampardoni[1,3], Lasya Sreepada[1,3], Dhivya Srinivasan[1,3], Junhao Wen[1,3], Zhijian Yang[1,2,100], Fanyang Yu[1,3], Sindhuja Govindarajan Tirumalai[1,3], Yuhan Cui[1,3], Zhen Zhou[1,3], Katharina Wittfeld[13,14], Hans J. Grabe[13,14], Duygun Tosun[15], Murat Bilgel[6], Yang An[6], Susan M. Resnick[6], John C. Morris[10], Daniel S. Marcus[16],

Pamela LaMontagne[16], Susan R. Heckbert[17], Thomas R. Austin[17], Lenore J. Launer[18], Aristeidis Sotiras[19], Mark Espeland[20], Colin L. Masters[8], Paul Maruff[8], Jurgen Fripp[9], Henry Völzk[21], Sterling C. Johnson[22], Marilyn S. Albert[7], Luigi Ferrucci[23] & R. Nick Bryan[24]

[13]German Center for Neurodegenerative Diseases (DZNE), Site Rostock/Greifswald, Germany. [14]Department of Psychiatry and Psychotherapy, University Medicine Greifswald, Greifswald, Germany. [15]Department of Radiology and Biomedical Imaging, University of California, San Francisco, San Francisco, CA, USA. [16]Department of Radiology, Washington University School of Medicine, St. Louis, MO, USA. [17]Department of Epidemiology, University of Washington, Seattle, WA, USA. [18]Intramural Research Program, Laboratory of Epidemiology and Population Sciences, National Institute on Aging, National Institutes of Health, Bethesda, MD, USA. [19]Mallinckrodt Institute of Radiology, Institute for Informatics, Washington University in Saint Louis, Saint Louis, MO, USA. [20]Department of Internal Medicine, Wake Forest School of Medicine, Winston-Salem, NC, USA. [21]Institute for Community Medicine, University Medicine Greifswald, 17475 Greifswald, Germany. [22]Wisconsin Alzheimer's Institute, University of Wisconsin School of Medicine and Public Health, Madison, WI, USA. [23]Longitudinal Study Section, Translational Gerontology Branch, National Institute on Aging, Baltimore, MD, USA. [24]Department of Diagnostic Medicine, University of Texas-Austin, Austin, TX, USA.

## Baltimore Longitudinal Study of Aging (BLSA)

Susan M. Resnick[6], Murat Bilgel[6], Yang An[6] & Luigi Ferrucci[23]

## Alzheimer's Disease Neuroimaging Initiative (ADNI)

Michael Weiner[25], Paul Aisen[26], Ronald Petersen[27], Clifford R. Jack Jr[27], William Jagust[28], John Q. Trojanowki[29], Arthur W. Toga[30], Laurel Beckett[31], Robert C. Green[32], Andrew J. Saykin[33], John C. Morris[10], Leslie M. Shaw[29], Enchi Liu[34], Tom Montine[35], Ronald G. Thomas[26], Michael Donohue[26], Sarah Walter[26], Devon Gessert[26], Tamie Sather[26], Gus Jiminez[26], Danielle Harvey[31], Matthew Bernstein[26], Nick Fox[36], Paul Thompson[37], Norbert Schuff[38], Charles DeCArli[31], Bret Borowski[27], Jeff Gunter[27], Matt Senjem[27], Prashanthi Vemuri[27], David Jones[27], Kejal Kantarci[27], Chad Ward[27], Robert A. Koeppe[39], Norm Foster[40], Eric M. Reiman[41], Kewei Chen[41], Chet Mathis[42], Susan Landau[28], Nigel J. Cairns[43], Erin Householder[30], Lisa Taylor Reinwald[43], Virginia Lee[44], Magdalena Korecka[44], Michal Figurski[44], Karen Crawford[30], Scott Neu[30], Tatiana M. Foroud[33], Steven Potkin[45], Li Shen[33], Faber Kelley[33], Sungeun Kim[33], Kwangsik Nho[33], Zaven Kachaturian[46], Richard Frank[47], Peter J. Snyder[48], Susan Molchan[49], Jeffrey Kaye[50], Joseph Quinn[50], Betty Lind[50], Raina Carter[50], Sara Dolen[50], Lon S. Schneider[30], Sonia Pawluczyk[30], Mauricio Beccera[30], Liberty Teodoro[30], Bryan M. Spann[30], James Brewer[26], Helen Vanderswag[26], Adam Fleisher[26], Judith L. Heidebrink[39], Joanne L. Lord[39], Ronald Petersen[27], Sara S. Mason[27], Colleen S. Albers[27], David Knopman[27], Kris Johnson[27], Rachelle S. Doody[51], Javier Villanueva Meyer[51], Munir Chowdhury[51], Susan Rountree[51], Mimi Dang[51], Yaakov Stern[52], Lawrence S. Honig[52], Karen L. Bell[52], Beau Ances[43], Maria Carroll[43], Sue Leon[43], Erin Householder[43], Mark A. Mintun[43], Stacy Schneider[43], Angela OliverNG[53], Randall Griffith[53], David Clark[53], David Geldmacher[53], John Brockington[53], Erik Roberson[53], Hillel Grossman[54], Effie Mitsis[54], Leyla deToledo-Morrell[55], Raj C. Shah[55], Ranjan Duara[56], Daniel Varon[56], Maria T. Greig[56], Peggy Roberts[56], Marilyn Albert[57], Chiadi Onyike[57], Daniel D'Agostino II[57], Stephanie Kielb[57], James E. Galvin[58], Dana M. Pogorelec[58], Brittany Cerbone[58], Christina A. Michel[58], Henry Rusinek[58], Mony J. de Leon[58], Lidia Glodzik[58], Susan De Santi[58], P. Murali Doraiswamy[59], Jeffrey R. Petrella[59], Terence Z. Wong[59], Steven E. Arnold[29], Jason H. Karlawish[29], David A. Wolk[1,11,12], Charles D. Smith[60], Greg Jicha[60], Peter Hardy[60], Partha Sinha[60], Elizabeth Oates[60], Gary Conrad[60], Oscar L. Lopez[42], MaryAnn Oakley[42], Donna M. Simpson[42], Anton P. Porsteinsson[61], Bonnie S. Goldstein[61], Kim Martin[61], Kelly M. Makino[61], M. Saleem Ismail[61], Connie Brand[61], Ruth A. Mulnard[45], Gaby Thai[45], Catherine Mc Adams Ortiz[45], Kyle Womack[62], Dana Mathews[62], Mary Quiceno[62], Ramon Diaz Arrastia[62], Richard King[62], Myron Weiner[62], Kristen Martin Cook[62], Michael DeVous[62], Allan I. Levey[63], James J. Lah[63], Janet S. Cellar[63], Jeffrey M. Burns[64], Heather S. Anderson[64], Russell H. Swerdlow[64], Liana Apostolova[65], Kathleen Tingus[65], Ellen Woo[65], Daniel H. S. Silverman[65], Po H. Lu[65], George Bartzokis[65], Neill R. Graff Radford[66], Francine ParfittH[66],

Tracy Kendall[66], Heather Johnson[66], Martin R. Farlow[33], Ann Marie Hake[33], Brandy R. Matthews[33], Scott Herring[33], Cynthia Hunt[33], Christopher H. van Dyck[67], Richard E. Carson[67], Martha G. MacAvoy[67], Howard Chertkow[68], Howard Bergman[68], Chris Hosein[68], Sandra Black[69], Bojana Stefanovic[69], Curtis Caldwell[69], Ging Yuek Robin Hsiung[70], Howard Feldman[70], Benita Mudge[70], Michele Assaly Past[70], Andrew Kertesz[71], John Rogers[71], Dick Trost[71], Charles Bernick[72], Donna Munic[72], Diana Kerwin[73], Marek Marsel Mesulam[73], Kristine Lipowski[73], Chuang Kuo Wu[73], Nancy Johnson[73], Carl Sadowsky[74], Walter Martinez[74], Teresa Villena[74], Raymond Scott Turner[75], Kathleen Johnson[75], Brigid Reynolds[75], Reisa A. Sperling[32], Keith A. Johnson[32], Gad Marshall[32], Meghan Frey[32], Jerome Yesavage[76], Joy L. Taylor[76], Barton Lane[76], Allyson Rosen[76], Jared Tinklenberg[76], Marwan N. Sabbagh[77], Christine M. Belden[77], Sandra A. Jacobson[77], Sherye A. Sirrel[77], Neil Kowall[78], Ronald Killiany[78], Andrew E. Budson[78], Alexander Norbash[78], Patricia Lynn Johnson[78], Thomas O. Obisesan[79], Saba Wolday[79], Joanne Allard[79], Alan Lerner[80], Paula Ogrocki[80], Leon Hudson[80], Evan Fletcher[81], Owen Carmichael[81], John Olichney[81], Charles DeCarli[81], Smita Kittur[82], Michael Borrie[83], T. Y. Lee[83], Rob Bartha[83], Sterling Johnson[84], Sanjay Asthana[84], Cynthia M. Carlsson[84], Steven G. Potkin[85], Adrian Preda[85], Dana Nguyen[85], Pierre Tariot[41], Adam Fleisher[41], Stephanie Reeder[41], Vernice Bates[86], Horacio Capote[86], Michelle Rainka[86], Douglas W. Scharre[87], Maria Kataki[87], Anahita Adeli[87], Earl A. Zimmerman[88], Dzintra Celmins[88], Alice D. Brown[88], Godfrey D. Pearlson[89], Karen Blank[89], Karen Anderson[89], Robert B. Santulli[90], Tamar J. Kitzmiller[90], Eben S. Schwartz[90], Kaycee M. SinkS[91], Jeff D. Williamson[91], Pradeep Garg[91], Franklin Watkins[91], Brian R. Ott[92], Henry Querfurth[92], Geoffrey Tremont[92], Stephen Salloway[93], Paul Malloy[93], Stephen Correia[93], Howard J. Rosen[25], Bruce L. Miller[25], Jacobo Mintzer[94], Kenneth Spicer[94], David Bachman[94], Elizabether Finger[95], Stephen Pasternak[95], Irina Rachinsky[95], John Rogers[95], Andrew Kertesz[95], Dick Drost[95], Nunzio Pomara[96], Raymundo Hernando[96], Antero Sarrael[96], Susan K. Schultz[97], Laura L. Boles Ponto[97], Hyungsub Shim[97], Karen Elizabeth Smith[97], Norman Relkin[98], Gloria Chaing[98], Lisa Raudin[98], Amanda Smith[99], Kristin Fargher[99] & Balebail Ashok Raj[99]

[25]UC San Francisco, San Francisco, CA, USA. [26]University of California San Diego, San Diego, CA, USA. [27]Mayo Clinic, Rochester, NY, USA. [28]UC Berkeley, Berkeley, CA, USA. [29]University of Pennsylvania, Philadelphia, PA, USA. [30]University of Southern California, Los Angeles, CA, USA. [31]UC Davis, Davis, CA, USA. [32]Brigham and Women's Hospital, Harvard Medical School, Boston, MA, USA. [33]Indiana University, Bloomington, IND, USA. [34]Janssen Alzheimer Immunotherapy, South San Francisco, CA, USA. [35]University of Washington, Seattle, WA, USA. [36]University of London, London, UK. [37]USC School of Medicine, Los Angeles, CA, USA. [38]UCSF MRI, San Francisco, CA, USA. [39]University of Michigan, Ann Arbor, MI, USA. [40]University of Utah, Salt Lake City, UT, USA. [41]Banner Alzheimer's Institute, Phoenix, AZ, USA. [42]University of Pittsburgh, Pittsburgh, PA, USA. [43]Washington University St. Louis, St. Louis, MO, USA. [44]UPenn School of Medicine, Philadelphia, PA, USA. [45]University of California, Irvine, CA, USA. [46]Khachaturian, Radebaugh & Associates, Inc and Alzheimer's Association's Ronald and Nancy Reagan's Research Institute, Chicago, IL, USA. [47]General Electric, Boston, MA, USA. [48]Brown University, Providence, RI, USA. [49]National Institute on Aging/National Institutes of Health, Bethesda, MD, USA. [50]Oregon Health and Science University, Portland, OR, USA. [51]Baylor College of Medicine, Houston, TX, USA. [52]Columbia University Medical Center, New York, NY, USA. [53]University of Alabama Birmingham, Birmingham, MO, USA. [54]Mount Sinai School of Medicine, New York, NY, USA. [55]Rush University Medical Center, Chicago, IL, USA. [56]Wien Center, Vienna, Austria. [57]Johns Hopkins University, Baltimore, MD, USA. [58]New York University, New York, NY, USA. [59]Duke University Medical Center, Durham, NC, USA. [60]University of Kentucky, Lexington, KY, USA. [61]University of Rochester Medical Center, Rochester, NY, USA. [62]University of Texas Southwestern Medical School, Dallas, TX, USA. [63]Emory University, Atlanta, GA, USA. [64]University of Kansas, Medical Center, Lawrence, KS, USA. [65]University of California, Los Angeles, CA USA. [66]Mayo Clinic, Jacksonville, FL, USA. [67]Yale University School of Medicine, New Haven, CT, USA. [68]McGill Univ., Montreal Jewish General Hospital, Montreal, WI, USA. [69]Sunnybrook Health Sciences, Toronto, ON, Canada. [70]U.B.C. Clinic for AD & Related Disorders, British Columbia, BC, Canada. [71]Cognitive Neurology St. Joseph's, Toronto, ON, Canada. [72]Cleveland Clinic Lou Ruvo Center for Brain Health, Las Vegas, NV, USA. [73]Northwestern University, Evanston, IL, USA. [74]Premiere Research Inst Palm Beach Neurology, West Palm Beach, FL, USA. [75]Georgetown University Medical Center, Washington, DC, USA. [76]Stanford University, Santa Clara County, CA, USA. [77]Banner Sun Health Research Institute, Sun City, AZ, USA. [78]Boston University, Boston, MA, USA. [79]Howard University, Washington, DC, USA. [80]Case Western Reserve University, Cleveland, OH, USA. [81]University of California, Davis Sacramento, CA, USA. [82]Neurological Care of CNY, New York, NY, USA. [83]Parkwood Hospital, Parkwood, CA, USA. [84]University of Wisconsin, Madison, WI, USA. [85]University of California, Irvine BIC, Irvine, CA, USA. [86]Dent Neurologic Institute, Amherst, MA, USA. [87]Ohio State University, Columbus, OH, USA. [88]Albany Medical College, Albany, NY, USA. [89]Hartford Hosp, Olin Neuropsychiatry Research Center, Hartford, CT, USA. [90]Dartmouth Hitchcock Medical Center, Albany, NY, USA. [91]Wake Forest University Health Sciences, Winston-Salem, NC, USA. [92]Rhode Island Hospital, Providence, RI, USA. [93]Butler Hospital, Providence, RI, USA. [94]Medical University South Carolina, Charleston, SC, USA. [95]St. Joseph's Health Care, Toronto, ON, Canada. [96]Nathan Kline Institute, Orangeburg, SC, USA. [97]University of Iowa College of Medicine, Iowa City, IA, USA. [98]Cornell University, Ithaca, NY, USA. [99]University of South Florida: USF Health Byrd Alzheimer's Institute, Tampa, FL, USA.

