## [Peer Review File · Nature Communications]

A deep learning framework identifies dimensional representations of Alzheimer's Disease from brain structureREVIEWER COMMENTS

Reviewer #1 (Remarks to the Author):

Yang et al. propose a clustering technique (Smile-GAN) based on a generative adversarial network. The authors apply Smile-GAN to a dataset comprising individuals from ADNI (CN/MCI/Dementia) and BLSA (almost exclusively CN), identifying four atrophy patterns - P1 (normal), P2 (mild/diffuse atrophy), P3 (medial temporal atrophy) and P4 (advanced neurodegeneration). Yang et al. then use longitudinal data to identify pathways between these different clusters, identifying two trajectories: P1->P2->P4 and P1->P3->P4.

Smile-GAN is technically well thought out with the key strengths being the formulation capturing non-disease related variations and the use of a generative adversarial network, which should allow clustering of voxel-wise data and thus identification of new subtype-specific features. The limitations of the work are that the authors apply Smile-GAN only to regions of interest and thus are unable to identify the new more sensitive features the GAN formulation should enable, and that the AD clusters identified by Smile-GAN seem to miss clusters of AD that have been identified previously.

Some more detailed comments:

1. The clustering results are not externally validated. At the very least it would be beneficial to show the same clustering results on ADNI1 and ADNI2 separately.
2. In the introduction and discussion the authors incorrectly assert that the lack of disease stage assumptions in Smile-GAN allows the method to indirectly capture mixed pathologies, in contrast with the SuStaIn algorithm. However, the presence or absence of disease stage assumptions does not affect the ability of an algorithm to model mixed pathology. Both SuStaIn and Smile GAN can be used post-hoc to capture mixed pathology by modelling an individual as a combination of subtypes. The authors also fail to mention that there are subtyping techniques that do deal directly with mixed pathology, such as Zhang et al. PNAS 2016.
3. The discussion needs referencing throughout - there are no references for LATE or the A/T/N framework in particular.
4. It would be beneficial to include a paragraph in the Discussion comparing the results of this study with other subtyping studies. Of particular note is the lack of a limbic-predominant atrophy pattern, which has been identified at autopsy and in numerous structural MRI-based clustering studies.

Reviewer #2 (Remarks to the Author):

Yang, Nasrallah, and colleagues evaluated 8,146 MRI scans, where they described and validated a semi-supervised deep clustering method capable of identifying morphological patterns associated with specific cognitive decline profiles. Also, they identified different longitudinal morphological patterns of progression and suggested that these patterns predict clinical progression better/complementary compared to amyloid and tau.

This is a well-powered and interesting study carried out with high methodological standards. The results have a potential impact on the field and the method has potential applications for a broader use. However, I have some concerns.

There are multiple different analyses in different figures that likely use different subsets of the data. For example, only a subset has CSF markers. It would be very helpful if the authors describe in the legend of the figures the subset used in each analysis and the number of subjects used in each analysis.

Could the authors briefly describe inclusion/exclusion criteria as well as diagnostic criteria of the different cohorts?

Is it possible to separate the result of the cross-sectional and longitudinal patterns in figure 1?

They are complementary but different information.

Figure 3b is difficult to understand.

In Figures 4a and b, why the authors do not use a 4-color scheme divided into 4 groups like one color for P1-P2 CN, another color for P-P2 MCI/dementia. The way it is it is not very intuitive.

Although one can infer, it is not clear in Figure 5a and b the p-value of hazard is in relation to which (control) group? Could the authors give more details in the legend? Are these box whisker plots?

Could the authors be clearer regarding statistical significance differences between groups in figure 5 c, d?

Again, in figure 6a, the results are statically significant different? Also, the authors should show "P" alone in this figure.

Figure 7 could be simplified.

Table 1. The authors should describe the presence of available CSF abeta/p-tau per clinical group.

Could the authors mention the MRI sequence they are using in the abstract

I could not find a description of the method used to CSF quantification in ADNI?

Could the authors give a little bit more details about the ADNI composite scores for memory, executive function, and language?

Reviewer #3 (Remarks to the Author):

The major claim of the paper is that a novel analysis of 145 brain-MRI volumes among 2 cohorts (ADNI and BLSA) allows characterizing qualitatively the atrophy of the brain associated with cognitive impairment. Specifically, the authors characterize 4 patterns. P1 corresponds to normal anatomy and P4 to advanced neurodegeneration. The patterns P2 and P3 are the more subtle corresponding respectively to mild diffuse atrophy and medial temporal atrophy, excluding other atrophy.

The main method allows discovering these patterns using a novel deep learning algorithm that builds upon Generative Adversarial Networks. The goal is to learn a mapping from the 145 brain-MRI volumes of a cognitively normal (CN) subject to the corresponding volumes of a cognitively impaired (CI) subject. While there are many such mappings that are consistent with the observed data, the authors impose further constraints on this mapping. Specifically, this mapping is constrained to be indexed by a probability vector, the probability to belong to each of the 4 subtypes P1 To P4. The smoothness of the mapping is also enforced.

Comments:

C1: My main concern is that the method is arguably complicated and still lacks statistical verification. I would suggest a permutation experiment where the CN and CI status of the subjects are shuffled and the learning is repeated. This experiment would help to assess if the findings are significant and compute p-values.

Other comments:

C2: Z is Uniformly distributed. Does this imply that the subtypes need to be uniformly distributed? If this is the case, even approximately, does it means that the method would not be appropriate to identify rare patterns?

C3: In the supplementary material, line 15, why the Lipschitz constant of g is the same as the Lipschitz constant of f? Isn't an artificial constraint?

C4: In the supplementary material, line 61: "We ensure Lipschitz continuity of functions f and g by performing weight clipping 2". However, in Ref 2, Arjovsky, M., Chintala, S. & Bottou, L. Wasserstein GAN. arXiv:1701.07875 (2017), I read "Weight clipping is a clearly terrible way to enforce a Lipschitz constraint." The authors need to explain why the authors of Ref. 2 are not correct in this case.

C5: Line 581: We used the Adjusted Rand Index (ARI)[37]. In the reference cited: Hubert, L. & Arabie, P. Comparing Partitions. Journal of Classification 2, 193–218 (1985), the author is critical of this index. Instead, the author proposes another index that is corrected for chance. Why, then,

the authors still use the ARI?

Dear Editor and Reviewers:

We are grateful to the reviewers for their constructive comments, which have been very helpful in revising our manuscript and in improving its quality.

We addressed the questions raised by the reviewers as follows. Main changes and corrections are marked in red in the revised manuscript.

Reviewer #1 comments and responses:

1. The limitations of the work are that the authors apply Smile-GAN only to regions of interest and thus are unable to identify the new more sensitive features the GAN formulation should enable, and that the AD clusters identified by Smile-GAN seem to miss clusters of AD that have been identified previously.

Response: We thank the reviewer for these comments. Machine learning models have been proved to be effective not only with voxel-level data, but also with features at varying resolutions, including that *a priori* defined ROIs. For instance, Wen et al. 2020 (doi:<https://doi.org/10.1016/j.media.2020.101694>) has objectively and reproducibly compared the classification performance of AD with different features, and proved that 3D features (e.g., voxel) obtained comparable accuracy with that of ROIs, using convolutional neural networks. That being said, Smile-GAN does have the potential for the extension to voxel-wise features, which is conceptually straight-forward. However, several challenges exist for such an implementation. First, training a GAN at the dimensionality of voxels is challenging from the perspectives of computational power, model stability (e.g, mode collapse) or over-fitting. Designed for lower dimensional ROI data, Smile-GAN does not require large computational power and can be a tool which is much more easily accessible for experiment and reproduction. Second, deriving multi-site harmonized data is crucial to define disease-related subtypes. In the current approach, we carefully teased out the site-effect from the ROI data (from the ADNI and BLSA studies) using a previously validated method, termed neuroHarmonize. (Pomponio et al., 2020, <https://doi.org/10.1016/j.neuroimage.2019.116450>, derived using ROIs from N=10477 scans across 18 studies over the entire life span). Multi-site or multi-scanner harmonization methods at ROI levels have been relatively well-explored in the literature (see another method called Neuroharmony: <https://doi.org/10.1016/j.neuroimage.2020.117127>). Voxel-wise harmonization is still under development and evaluation, and certainly an important future direction of our work. Our preliminary efforts are in line with the extension of Smile-GAN (<https://arxiv.org/pdf/2010.05355.pdf>). Lastly, a recent study (SuStaIn, Young et al¹) has shown success using fewer ROIs. The dimension of our MUSE ROIs (N=145) is relatively finer and has been shown to be statistically sufficient for Smile-GANs to learn its underlying distribution, at this scale. Of course, it would be quite feasible to apply Smile-GAN with higher dimensional features, e.g. >1000 components from data-driven parcellation methods, such as orthogonally-projective NMF ([10.1016/j.neuroimage.2014.11.045](https://doi.org/10.1016/j.neuroimage.2014.11.045)), which may allow for more refined pattern generation. Our current study serves to establish this potential for Smile-GAN.

Understanding the relationship to validated subtypes of AD is important for understanding the output of Smile-GAN. We have revised the manuscript to include reference to imaging biomarkers for pathologically-defined subtypes of AD, specifically hippocampal sparing and limbic predominant subtypes², showing data that suggests that a hippocampal sparing atrophy

pattern corresponds to P2 and a limbic predominant atrophy pattern corresponds to P3 (Supplemental Fig 4). The 4 subtypes defined by Smile-GAN correspond well to the four subtypes found by another semi-supervised method, termed CHIMERA³, using a similar population (AD+MCI from ADNI). Smile-GAN patterns in large part correspond to subtypes identified by recent unsupervised methods, although variations exist due to different choices in population or methods, etc. For instance, Zhang et al.⁴, PNAS, 2016 specifically focused on late-onset AD and found 3 subtypes (temporal dominant subtype akin to P3, typical widespread cortical subtype akin to P2 or P4, and a subcortical subtype). Smile-GAN did not identify a distinct subcortical subtype, and we address potential reasons for this in the revised manuscript (Page 15, paragraph 2; Supplemental section 1.7 and 2.5). We interpret SmileGAN's results as consistent with the known neuropathological patterns that have been established via post-mortem analyses in AD, although our sample does not have neuropathologic correlation. However, more studies need to be conducted to specifically elucidate whether the apparent subcortical subtype is disease-related or perhaps a neuroanatomical variation unrelated to AD pathology. Besides some similarities and some inconsistencies in results, the method Smile-GAN itself is very different from many published methods since it aims to only derive patterns related to disease effects via a novel deep learning framework and it also shows superiority over other semi-supervised clustering methods as shown in supplementary experiments with known ground truth.

2. The clustering results are not externally validated. At the very least it would be beneficial to show the same clustering results on ADNI1 and ADNI2 separately.

Response: We would like to thank the reviewer for this suggestion. We failed to clearly convey that we actually did train the model on ADNI2 data and test it on ADNI1 and BLSA data. In the revised manuscript, we added 3 more experiments to verify the reproducibility and generalizability of our findings: i) train on ADNI2/GO and test on ADNI1 data; ii) retrain the model on ADNI1 and test on ADNI 1; iii) train and test the model on a larger iSTAGING dataset including data harmonized across seven different studies. VBM results of all three different experiments show patterns similar to what we derived based on training from the ADNI2/GO dataset only. (Supplementary section 1.5 and 2.2)

3. In the introduction and discussion the authors incorrectly assert that the lack of disease stage assumptions in Smile-GAN allows the method to indirectly capture mixed pathologies, in contrast with the SuStaIn algorithm. However, the presence or absence of disease stage assumptions does not affect the ability of an algorithm to model mixed pathology. Both SuStaIn and Smile GAN can be used post-hoc to capture mixed pathology by modelling an individual as a combination of subtypes. The authors also fail to mention that there are subtyping techniques that do deal directly with mixed pathology, such as Zhang et al. PNAS 2016.

Response: We appreciate this fair point. We might have conveyed the wrong message originally, however what we meant is that Smile-GAN can still capture stage and longitudinal change via a second level modeling of the trajectories of memberships to these main patterns. We consider our 4 patterns as a dimensional system within which longitudinal trajectories can be explored. We have now reworded our introduction and discussion and included more introduction and discussion on previous works like Young et al¹. and Zhang et al⁴.

4. The discussion needs referencing throughout - there are no references for LATE or the A/T/N framework in particular.

Response: We agree that these references are important. We have added references for LATE and A/T/N in the revised manuscript.

5. It would be beneficial to include a paragraph in the Discussion comparing the results of this study with other subtyping studies. Of particular note is the lack of a limbic-predominant atrophy pattern, which has been identified at autopsy and in numerous structural MRI-based clustering studies.

Response: We have included one paragraph of discussion on the relationship between results of other subtyping studies and our results. We also implemented further experiments to explore subtypes presented in previous works, including Limbic Predominant, Hippocampal Sparing, typical AD and subcortical types. Detailed experiments and results can be found in Supplementary Method section 1.7 and Supplementary Result section 2.5.

The discussion paragraph is copied here:

This four pattern system has similarities with other neuroimaging-based clustering studies, including identification of temporal and cortical predominant patterns.^{1,3-5} Smile-GAN patterns tentatively correspond to pathologically-identified subtypes of AD²: Limbic Predominant, matching P3, Hippocampal Sparing, matching P2, and typical AD (mixed P2-P3). There is one another possible subtype of subcortical atrophy previously identified by several other MRI-based unsupervised clustering algorithms using ADNI data^{4,6} but not distinctly identified by Smile-GAN. There are a few reasons why smile-GAN did not identify a subcortical pattern. First, as observed in Zhang et al⁷, the subcortical pattern may merge with the temporal pattern based upon harmonization and clustering methodology and specific training sample. Second, a portion of the variability attributed to a subcortical atrophy may not be disease-related, resulting in insufficient signal to separately cluster as a distinct pattern, a possibility potentially supported by the lack of pathological evidence for this subtype and scarce atrophy in subcortical ROIs among the patient group (Supplementary Fig. 5).

Reviewer #2 comments and responses:

1. There are multiple different analyses in different figures that likely use different subsets of the data. For example, only a subset has CSF markers. It would be very helpful if the authors describe in the legend of the figures the subset used in each analysis and the number of subjects used in each analysis. Could the authors briefly describe inclusion/exclusion criteria as well as diagnostic criteria of the different cohorts?

Response: We thank the reviewer for pointing out this important point that was not clearly detailed in the initial manuscript. We have now included details of subset and number of subjects in the legend of each figure. We included all datapoints available from ADNI1/2/GO and BLSA study, though different subsets of data are selected for different analysis depending on specific requirement, with the major narrowing of the data set relating to the availability of amyloid/tau measures only in the ADNI sample. Details of the subsets used for each figure can now be found in figure legends and in the Method section. We have also included references to diagnostic criteria for ADNI and BLSA in method section 4.2. Briefly, the BLSA enrolls participants who live within 2 hours of Baltimore, MD who are age 20+ and in good health and without cognitive or functional impairments at baseline, and follows them longitudinally, with a subset having neuroimaging. ADNI recruits at sites across the US and Canada, enrolling MCI participants (defined as CDR=0.5 with memory complaints but largely intact cognition and functional performance) and AD participants (CDR=0.5-1 with memory complaints and mild disease severity meeting NINCDS-ADRDA criteria for probable AD). ADNI controls were age-matched to MCI/AD participants, without memory complaints, and CDR=0. More detailed information of enrollment criteria can be found in Resnick et al.⁸ (<https://www.ncbi.nlm.nih.gov/pmc/articles/PMC6742337/> and the website: <https://www.nia.nih.gov/research/labs/blsa/join-blsa>) for BLSA and Peterson et al.⁹ (<https://www.ncbi.nlm.nih.gov/pmc/articles/PMC2809036/>) for ADNI.

2. Is it possible to separate the result of the cross-sectional and longitudinal patterns in figure 1? They are complementary but different information.

Response: Based on the question, we infer that the reviewer is asking about the first data figure, which is Figure 2. We have separated Figure 2 into top (a,b) and bottom (c,d) parts for cross-sectional and longitudinal information respectively. We agree that they are complementary information.

3. Figure 3b is difficult to understand.

Response: We have included more explanation for this figure and changed the color to make it more visible.

Additional text in Results section 2.3:

In Fig. 3b, participants are grouped as normal, as falling along the typical AD continuum, as AD with dominant copathology or as suspected non-AD pathology (SNAP) based on patterns and Abeta/phospho-tau (pTau) status. A+T+ participants tend to have more severe neurodegeneration than A+T- participants,

as expected. Using pattern membership as a classification of (N) modestly increases the number of classes (from 8 to 16) but provides important severity and prognostic information.

4. In Figures 4a and b, why the authors do not use a 4-color scheme divided into 4 groups like one color for P1-P2 CN, another color for P-P2 MCI/dementia. The way it is not very intuitive.

Response: We used 2 colors to make it easier to compare participants progressing along the same path but understand that this choice may make the figure unintuitive. We have changed the figure and used two different but similar colors for participants progressing along the same path.

5. Although one can infer, it is not clear in Figure 5a and b the p-value of hazard is in relation to which (control) group? Could the authors give more details in the legend? Are these box whisker plots?

Response: In Fig. 5a and b, we show p-values of hazards of A+/T+ patients in relation to those with A- or T- status and Fig 5c/d are box and whisker plots. We have updated the legend to make this information clearer.

6. Could the authors be clearer regarding statistical significance differences between groups in figure 5 c, d?

Response: We have run one-way ANOVA to test the significance of differences between different groups of CIs derived using different biomarkers. Detailed p values can be found in section 2.5 “Prediction of Clinical Progression (Change in Diagnosis).”

7. Again, in figure 6a, the results are statistically significantly different? Also, the authors should show “P” alone in this figure.

Response: We have run same statistical tests mentioned above and provided p values in section 2.5 “Composite Score for Risk of Clinical Progression”. Also, we have added “P” alone in figure 6a for better comparisons.

8. Figure 7 could be simplified.

Response: We have updated figure 7 to a more simplified version. In the new version, we have removed the number of participants in ADNI at each node and have made the thickness of node connections to represent approximate flux through a node. The purpose here is to allow the reader to see the major biomarker pathways and the minor means of switching between pathways.

9. Table 1. The authors should describe the presence of available CSF abeta/p-tau per clinical group.

Response: The availability of CSF abeta/ptau per clinical group has been added in Method section 4.5. We have also added information about samples used for each analysis in figure legends.

10. Could the authors mention the MRI sequence they are using in the abstract?

Response: We have included the MRI sequence (3D isotropic T1) in the abstract.

11. I could not find a description of the method used to CSF quantification in ADNI?

Response: We have added reference to the method used for CSF quantification in Method section 4.5.

12. Could the authors give a little bit more details about the ADNI composite scores for memory, executive function, and language?

Response: We have included more details on ADNI composite scores in Method section 4.5, with delineation of the tests used within each composite. The derivation of these composite scores was complex and we have also included a reference to a thorough description on the ADNI website as well as key references for further details.

Reviewer #3 comments and responses:

1. My main concern is that the method is arguably complicated and still lacks statistical verification. I would suggest a permutation experiment where the CN and CI status of the subjects are shuffled, and the learning is repeated. This experiment would help to assess if the findings are significant and compute p-values.

Response: We thank the Reviewer for this constructive suggestion, which is an important test of the validity of the Smile-GAN model. Based on the Reviewer's suggestion, we have implemented the permutation experiment, which help validate the significance of reproducibility of four patterns we derived using the Smile-GAN model. Briefly, this experiment shows failure to generate patterns using a permuted data set and statistical significance of the patterns obtained from the non-permuted data. Detailed experiments and results can be found in Supplementary section 1.4 and 2.4. Moreover, we would like to emphasize that we have provided validation experiments with known ground truth, which further establish the validity of Smile-GAN (Supplementary Method 2.1).

2. Z is Uniformly distributed. Does this imply that the subtypes need to be uniformly distributed? If this is the case, even approximately, does it mean that the method would not be appropriate to identify rare patterns?

Response: We would like to thank the reviewer for raising this important issue. Theoretically, when different patterns are uniformly distributed, i.e., each cluster has the same amount of patients, the smile-GAN method would have optimal discriminative ability. To better understand the performance in the setting of rare subgroups, we implemented additional experiments to test robustness of Smile-GAN model to both unbalanced subgroups and non-uniform distribution of variable Z. (Supplemental section 1.3.3 and 2.1.3). Results reveal that a moderate imbalance of subgroups does not significantly affect the performance of the model, but a very scarce pattern ($\leq 10\%$ of the sample) does undermine the clustering accuracy. A possible explanation for model success with moderately imbalanced data sets might be that equality in distribution of generated data and real data is never really reached in real cases, but the model is just minimizing the J-S divergence between two distributions. Alternatively, the mini-batch optimization procedure might be another reason why the model can deal with somewhat imbalanced subgroups. However, the model is shown to be not very robust to a non-uniform distribution of Z, suggesting that our selection of a discrete uniform distribution for the Z variable is better for the current implementation of Smile-GAN. However, it is possible that the model could be improved in the future to better identify rare patterns. We have included the limitation that rare or subtle patterns of atrophy may not be distinctly learned by the Smile-GAN model.

3. In the supplementary material, line 15, why is the Lipschitz constant of g same as the Lipschitz constant of f ? Isn't an artificial constraint?

Response: We thank the reviewer for catching this error. Lipschitz constants of f and g are not necessarily the same. We have changed the notation to K_1 for Lipschitz constant of f and K_2 for Lipschitz constant of g in the supplementary. In application, weight clipping parameter was artificially set to one same number, but slight changes to either one of them does not affect the performance of the model.

4. In the supplementary material, line 61: “We ensure Lipschitz continuity of functions f and g by performing weight clipping 2”. However, in Ref 2, Arjovsky, M., Chintala, S. & Bottou, L. Wasserstein GAN. arXiv:1701.07875 (2017), I read “Weight clipping is a clearly terrible way to enforce a Lipschitz constraint.” The authors need to explain why the authors of Ref. 2 are not correct in this case.

Response: We also noticed the claim that weight clipping is not a good way to enforce a Lipschitz constraint, but it was still used in W-GAN as the way to enforce it. They considered weight clipping a terrible method mainly because of the difficulty in choosing the right value for the clipping bound. In their cases, if the clipping parameter is large, then it can be hard to train the critic till optimality. If the clipping is small, this can easily lead to vanishing gradients when the number of layers is big.

There are several differences in our application and theirs. First, they enforce Lipschitz constraint for function f which plays the role of ‘Discriminator’ for minimizing the dual form of Wasserstein distance, while we are enforcing Lipschitz constraint for mapping and clustering function in order to regularize the mapping directions. Thus, variations in clipping parameters lead to different kinds of influence on our results. Second, dealing with ROI data, our network is much smaller and shallower than what they used for W-GAN, so there is no significant convergence and vanishing gradients problems in our task. Third, the main task for our problem is clustering rather than data generation, and clustering performance is pretty robust to different choices of clipping parameters. Therefore, we consider the weight clipping method good enough for our model.

5. Line 581: We used the Adjusted Rand Index (ARI)[37]. In the reference cited: Hubert, L. & Arabie, P. Comparing Partitions. Journal of Classification 2, 193–218 (1985), the author is critical of this index. Instead, the author proposes another index that is corrected for chance. Why, then, the authors still use the ARI?

Response: We apologize for using a shorthand for the ARI version implemented in this study, which is the ARI_HA. This measure is indeed the measure recommended in Hubert and Arabie (1985) where the authors proposed one of the many versions of Rand index adjusted for chance and the paper is commonly cited for the Adjusted Rand Index (i.e., ARI_HA) (e.g., https://scikit-learn.org/stable/modules/generated/sklearn.metrics.adjusted_rand_score.html, or https://en.wikipedia.org/wiki/Rand_index#Adjusted_Rand_index).

Hubert and Arabie., 1985 proposed one of the many adjusted versions of Rand index and is the paper commonly cited for Adjusted Rand Index (i.e., ARI_HA) (e.g., https://scikit-learn.org/stable/modules/generated/sklearn.metrics.adjusted_rand_score.html, or https://en.wikipedia.org/wiki/Rand_index#Adjusted_Rand_index). From the content of the paper itself, the authors didn't criticize ARI, but proposed their adjusted version for the first time. Here is the story behind the ARI. The Rand index was firstly proposed by Rand [Rand = $(a+d) / (a+b+c+d)$], 1971 (<https://www.jstor.org/stable/2284239?seq=1>). Morey and Agresti., 1984 (<https://doi.org/10.1177/0013164484441003>) realized that the above-mentioned formulation may be overinflated due to chance assignment and proposed their correction index for chance (ARI_MA). Hubert and Arabie (1985) found something amiss with the unwieldy measure in ARI_MA; specifically, Morey and Agresti (1984) incorrectly assumed that the expectation of a squared random variable is the square of the expectation. Hubert and Arabie corrected the original rand index with the proper adjustment, creating the ARI_HA (please see paper: D. Steinley, Properties of the Hubert-Arabie adjusted Rand index, Psychological Methods 2004). We now added one sentence in section 4.6 to clarify the reason why we adopted ARI_HA.

Reference

- 1 Young, A. L. *et al.* Uncovering the heterogeneity and temporal complexity of neurodegenerative diseases with Subtype and Stage Inference. *Nature communications* **9**, 4273, doi:10.1038/s41467-018-05892-0 (2018).
- 2 Murray, M. E. *et al.* Neuropathologically defined subtypes of Alzheimer's disease with distinct clinical characteristics: a retrospective study. *Lancet Neurol* **10**, 785-796, doi:10.1016/S1474-4422(11)70156-9 (2011).
- 3 Dong, A. *et al.* Heterogeneity of neuroanatomical patterns in prodromal Alzheimer's disease: links to cognition, progression and biomarkers. *Brain : a journal of neurology* **140**, 735-747, doi:10.1093/brain/aww319 (2017).
- 4 Zhang, X. *et al.* Bayesian model reveals latent atrophy factors with dissociable cognitive trajectories in Alzheimer's disease. *Proceedings of the National Academy of Sciences of the United States of America* **113**, E6535-e6544, doi:10.1073/pnas.1611073113 (2016).
- 5 Risacher, S. L. *et al.* Alzheimer disease brain atrophy subtypes are associated with cognition and rate of decline. *Neurology* **89**, 2176-2186, doi:10.1212/wnl.0000000000004670 (2017).
- 6 Young, A. L. *et al.* Uncovering the heterogeneity and temporal complexity of neurodegenerative diseases with Subtype and Stage Inference. *Nat Commun* **9**, 4273, doi:10.1038/s41467-018-05892-0 (2018).
- 7 Al-Kadi, O. S. Texture measures combination for improved meningioma classification of histopathological images. *Pattern Recognition* **43**, 2043-2053 (2010).
- 8 Resnick, S. M., Pham, D. L., Kraut, M. A., Zonderman, A. B. & Davatzikos, C. Longitudinal Magnetic Resonance Imaging Studies of Older Adults: A Shrinking Brain. *The Journal of Neuroscience* **23**, 295-301 (2003).
- 9 Petersen, R. C. *et al.* Alzheimer's Disease Neuroimaging Initiative (ADNI): clinical characterization. *Neurology* **74**, 201-209, doi:10.1212/WNL.0b013e3181cb3e25 (2010).

REVIEWER COMMENTS

Reviewer #1 (Remarks to the Author):

In general, I am satisfied by the authors response. However, I have a few outstanding concerns related the response to the first comment (that a limitation of the work is that Smile-GAN is applied only to regions of interest and thus can't identify the more sensitive features that the GAN formulation should enable).

1. In their response the authors reference Wen et al. MedIA 2020 as an example that ROIs obtain similar performance to 3D features. However, the ROI-based CNNs referred to in the paper use image patches within an ROI as features. From the Methods it reads like Smile-GAN is applied to overall volumes of different ROIs rather than image patches within ROIs. As Wen et al. compare ROI-based CNNs that use image patches rather than the overall volume of the ROI, the performance of Smile-GAN could not be expected to be comparable. This point requires further clarification.

2. The wording of the article is quite vague in places, which I think has the potential to be misleading given that the expectation for a GAN formulation would be that the features used are portions of the images themselves. For example, terminology such as "synthesise realistic scans" could imply that the features are obtained directly from the image, and "neuroanatomical patterns" could refer to any data obtained from structural MRI. I suggest the following changes to make the text clearer.

a. Abstract: "When applied to isotropic 1.5T/3T MRIs" to "When applied to regional atrophy levels obtained from structural 1.5T/3T MRIs"

b. Introduction first paragraph: "leveraging a GAN that is trained to synthesize realistic scans that are hard to distinguish from real patient scans" to "leveraging a GAN that is trained to synthesize realistic levels of regional atrophy that are hard to distinguish from levels of regional atrophy in real patients"

c. Introduction fourth paragraph: "Herein, we used GANs to synthesize exceptionally realistic patient imaging data" to "Herein, we used GANs to synthesize exceptionally realistic levels of regional atrophy"

d. Introduction fifth paragraph: "which captures different disease-related neuroanatomical patterns by generating realistic data through transformation of neuroanatomical data of CN individuals" to "which captures different disease-related neuroanatomical patterns by generating realistic levels of regional atrophy through transformation of ROI-based neuroanatomical data of CN individuals"

e. Results first paragraph: "Experiments on the semi-synthetic dataset, a real MRI dataset with synthesized brain atrophy" to "Experiments on the semi-synthetic dataset, a dataset that synthesizes brain atrophy in different ROIs based on real MRI data"

f. Results second paragraph: "Smile-GAN identified four significantly reproducible and disease-related patterns of brain atrophy" to "Smile-GAN identified four significantly reproducible and disease-related patterns of regional brain atrophy"

g. Discussion first paragraph: "We have developed a new deep learning approach, Smile-GAN, which disentangles anatomical heterogeneity and defines subtypes of neurodegeneration by learning to generate mappings from images of cognitively normal individuals to images of patients." To "We have developed a new deep learning approach, Smile-GAN, which disentangles anatomical heterogeneity and defines subtypes of neurodegeneration by learning to generate mappings from regional atrophy levels of cognitively normal individuals to regional atrophy levels of patients."

h. Discussion third paragraph: "Application of Smile-GAN to MRI data from a sample enriched with AD pathology identified 4 patterns of brain atrophy" to "Application of Smile-GAN to MRI data from a sample enriched with AD pathology identified 4 patterns of regional brain atrophy"

i. Discussion seventh paragraph: "It captures biologically relevant atrophy patterns that are few in number" to "It captures biologically relevant regional atrophy patterns that are few in number".

3. The discussion of the use of ROI-based atrophy levels and the potential to use more fine-grained features in future implementations of Smile-GAN is important and should be included in the paper.

A few further clarifications it would be beneficial to include:

1. I couldn't find much detail on the k-means and gaussian mixture modelling (GMM) methods the authors compare Smile-GAN to. I had the following questions related to this point.

- a. Which implementations of k-means and GMM were used (for reproducibility)?
 - b. How many start points were used? Both methods are known to be sensitive to initialisation.
 - c. Were the data normalised before applying k-means and GMM?
 - d. Were k-means and GMM applied to just patients or patients and controls?
 - e. Why are the methods performing worse than chance in Suppl. Table 4? If there are three clusters the expected chance performance would be 1/3?
2. I may have missed it but I couldn't find any information on how Figure 7 was generated? Is it cross-sectional data or longitudinal? Is the figure based on the assumption that people will progress in a certain order through the stages?

Reviewer #3 (Remarks to the Author):

I would like to thank the authors for their work in submitting a new improved version of the manuscript.

Thank you for performing the permutation test. However, there are still elements in this procedure that are unsatisfactory. Most importantly, the number of permutations, 50, is too small. Indeed, a p-value is the probability of an extreme event and thus requires more repetitions. I would suggest 1000 repetitions.

Also, the text is too vague to understand how is the procedure done. I read "To do so, we utilized the whole discovery set (i.e., 297 CN and 602 MCI/Dementia participants from ADNI2/GO) and randomly shuffled the labels of 'CN' and 'PT' to create a null dataset with 297 new 'CN' and 602 'PT' whose distributions are comparable."

Please be specific on the method used. "Whose distribution is comparable" is vague. Next, it would be helpful to know what exactly is inside the permutation loop.

A description similar to Algorithm 1 would be helpful.

Also, I would recommend redoing this for various values of M, say, M=2,3,4,5.

Indeed, the number of clusters is a well-known difficult parameter to estimate and is critical in the manuscript.

Next, for improving readability, could you define the use of the word "Domain" in Fig 8? Would a simpler word like vector or long vector suffice? Is the dimension the number of ROI? Is it 145 for all experiments?

Also, "atrophy" is used extensively. Could it be defined early? Similarly for the ROIs. Are the white matter regions excluded? what about the ventricles?

Dear Editor and Reviewers:

We are grateful to the reviewers for their constructive comments, which have been very helpful in revising our manuscript and in improving its quality.

We addressed the questions raised by the reviewers as follows. The primary changes and corrections are marked in red in the revised manuscript.

Reviewer #1:

In general, I am satisfied by the authors response. However, I have a few outstanding concerns related the response to the first comment (that a limitation of the work is that Smile-GAN is applied only to regions of interest and thus can't identify the more sensitive features that the GAN formulation should enable).

1. In their response the authors reference Wen et al. MedIA 2020 as an example that ROIs obtain similar performance to 3D features. However, the ROI-based CNNs referred to in the paper use image patches within an ROI as features. From the Methods it reads like Smile-GAN is applied to overall volumes of different ROIs rather than image patches within ROIs. As Wen et al. compare ROI-based CNNs that use image patches rather than the overall volume of the ROI, the performance of Smile-GAN could not be expected to be comparable. This point requires further clarification.

Response: We apologize for the confusion led by this reference. ROI-based image patches (3D hippocampal patch or randomly cropped patches, or the whole 3D tissue maps) in Wen *et al.* are indeed different from ROI-based regional volumes. The message we want to convey is that features with different dimension and resolutions are shown to be comparably effective for machine learning algorithms. To this point, Wen *et al.* systematically compare performance of CNNs and linear SVMs with different type of voxel-based features (3D patches, 3D ROI, GM density maps etc.). In an earlier work from the same group, González *et al.* (doi: <https://doi.org/10.1016/j.neuroimage.2018.08.042>) specifically compare classification performance of SVM with voxel-based and regional-based features. These works show a comparable performance of classification algorithm with different types of features derived from structural MRI, including both ROI-based regional volumes from an anatomical atlas or high-dimensional voxel-wise maps.

However, with that being said, we do not deny that other more sensitive features, such as voxel-wise density maps or other data-driven regional features (e.g., opNMF: [10.1016/j.neuroimage.2014.11.045](https://doi.org/10.1016/j.neuroimage.2014.11.045)), could be applied to Smile-GAN and potentially result in better clustering performance, particularly for more heterogeneous samples. However, in order to fit multi-site voxel-wise maps to GANs, efforts need to be made to overcome unique challenges including voxel-wise image harmonization, GAN model stability, availability of computational resources, etc. We appreciate the reviewer for raising this point, which is currently a very active topic in the field. We have added a brief discussion to the limitations section (Discussion on Page 17) as follows:

“Finally, the Smile-GAN model is currently applied to ROI volume data derived from MRI images only, and thus may fail to capture more subtle patterns that do not conform to anatomic ROIs. The Smile-GAN model architecture is flexible for use with smaller ROI parcellations or voxel-based analyses as well as non-structural MRI and non-imaging data. Extension of current framework to such other types of data is a direction for future development.”

2. The wording of the article is quite vague in places, which I think has the potential to be misleading given that the expectation for a GAN formulation would be that the features used are portions of the images themselves. For example, terminology such as “synthesise realistic scans” could imply that the features are obtained directly from the image, and “neuroanatomical patterns” could refer to any data obtained from structural MRI. I suggest the following changes to make the text clearer.

a. Abstract: “When applied to isotropic 1.5T/3T MRIs” to “When applied to regional atrophy levels obtained from structural 1.5T/3T MRIs”

b. Introduction first paragraph: “leveraging a GAN that is trained to synthesize realistic scans that are hard to distinguish from real patient scans” to “leveraging a GAN that is trained to synthesize realistic levels of regional atrophy that are hard to distinguish from levels of regional atrophy in real patients”

c. Introduction fourth paragraph: “Herein, we used GANs to synthesize exceptionally realistic patient imaging data” to “Herein, we used GANs to synthesize exceptionally realistic levels of regional atrophy”

d. Introduction fifth paragraph: “which captures different disease-related neuroanatomical patterns by generating realistic data through transformation of neuroanatomical data of CN individuals” to “which captures different disease-related neuroanatomical patterns by generating realistic levels of regional atrophy through transformation of ROI-based neuroanatomical data of CN individuals”

e. Results first paragraph: “Experiments on the semi-synthetic dataset, a real MRI dataset with synthesized brain atrophy” to “Experiments on the semi-synthetic dataset, a dataset that synthesizes brain atrophy in different ROIs based on real MRI data”

f. Results second paragraph: “Smile-GAN identified four significantly reproducible and disease-related patterns of brain atrophy” to “Smile-GAN identified four significantly reproducible and disease-related patterns of regional brain atrophy”

g. Discussion first paragraph: “We have developed a new deep learning approach, Smile-GAN, which disentangles anatomical heterogeneity and defines subtypes of neurodegeneration by learning to generate mappings from images of cognitively normal individuals to images of patients.” To “We have developed a new deep learning approach, Smile-GAN, which disentangles anatomical heterogeneity and defines subtypes of neurodegeneration by learning to generate mappings from regional atrophy levels of cognitively normal individuals to regional atrophy levels of patients.”

h. Discussion third paragraph: “Application of Smile-GAN to MRI data from a sample enriched with AD pathology identified 4 patterns of brain atrophy” to “Application of Smile-GAN to MRI data from a sample enriched with AD pathology identified 4 patterns of regional brain atrophy”

i. Discussion seventh paragraph: “It captures biologically relevant atrophy patterns that are few in number” to “It captures biologically relevant regional atrophy patterns that are few in number”.

Response: We thank the reviewer for these excellent recommendations; we agree that they improve clarity. We have edited the manuscript as suggested.

3. The discussion of the use of ROI-based atrophy levels and the potential to use more fine-grained features in future implementations of Smile-GAN is important and should be included in the paper.

Response: We agree that applying the Smile-GAN model to ROI volume data may fail to capture more subtle patterns. As stated in the response to the Reviewer's point 1, we have included this as a limitation of this work (Discussion on Page 17). Extension of current framework to patch-based and voxel-based data is a future direction of this work.

A few further clarifications it would be beneficial to include:

1. I couldn't find much detail on the k-means and gaussian mixture modelling (GMM) methods the authors compare Smile-GAN to. I had the following questions related to this point.

Response: We appreciate the reviewer for raising the importance of transparency and reproducibility in our field. In the revised manuscript, we have added more details regarding the other clustering methods used for comparison, including version of online packages, and selections of hyper-parameters. (Supplemental Method 1.3.2 on Page 6). Detailed answers to the following questions can be found below and are addressed in changes to the supplemental materials (Supplemental Method 1.3.2 on Page 6 and Supplemental Table 4 on page 11):

a. Which implementations of k-means and GMM were used (for reproducibility)?

b. How many start points were used? Both methods are known to be sensitive to initialization.

(a)(b) For implementation of HYDRA, we used the online PyPI package 'pyHYRDA==1.0.8'. For Chimera, we used the GitHub package (<https://github.com/aoyandong/CHIMERA>). K-means and GMM were both implemented via the package 'scikit-learn==0.24.2' with 400 initial points and 500 max-iterations; convergence threshold for both K-means and GMM were set to be $1e-5$. Further increase of initial points and max-iterations and decrease of convergence thresholds do not improve the performance of models. For GMM, K-means was chosen as the method for weight initialization and covariance type was set to be 'full'. (Supplemental Page 6, Section 1.3.2)

c. Were the data normalized before applying k-means and GMM?

(c) For HYDRA and CHIMERA, training data were normalized as required for each model. For the K-means and GMM clustering methods, PT data were normalized with respect to CN data to ensure a mean of 0 and standard deviation of 1 among CN participants for each ROI.

d. Were k-means and GMM applied to just patients or patients and controls?

(d) Since different patterns of atrophy are simulated to only 'Pseudo-PT' data, K-means and GMM were applied to cluster patients only. However, PT data are normalized with respect to CN beforehand. Moreover, since K-means and GMM do not have equal access to CN data like the other three semi-supervised methods, we implemented one additional, separate variant of the K-means and GMM methods according to Dong et al. (doi: [10.1109/TMI.2015.2487423](https://doi.org/10.1109/TMI.2015.2487423)): we clustered using a computed "distance" for each PT participant, which is the difference vector between each PT and its Euclidean nearest neighbor in the CN group.

e. Why are the methods performing worse than chance in Suppl. Table 4? If there are three clusters the expected chance performance would be 1/3?

(e) We apologize for a coding error we made in the experiment for this table and thank the reviewer for catching it. Precisely speaking, the previous results of the accuracy of K-means and GMM were not correct. Both ground truth and cluster labels given by semi-supervised methods range from 1-3, but K-means and GMM offers a label from 0-2, resulting in a mismatch. We

have corrected these results, which now show better performance for K-means and GMM methods. Furthermore, we added two additional experiments with different variants of K-means and GMM methods for fairer comparisons to semi-supervised algorithms. However, Smile-GAN still outperforms all compared clustering methods.

2. I may have missed it but I couldn't find any information on how Figure 7 was generated? Is it cross-sectional data or longitudinal? Is the figure based on the assumption that people will progress in a certain order through the stages?

Response: We thank the reviewer for raising this question. This figure is not completely derived from data, though it is partially based on distribution of cross-sectional data with different A/T/P category (Fig3b). Rather, it is a hypothetical model demonstrating how Smile-GAN patterns and A/T phenotypes diverge or converge in the ADNI sample. Transformations can not be directly derived from data because of the insufficient longitudinal CSF measures, so they are indeed based on the assumption that events happen following certain orders. (A-→A+; T-→T+; P1→P2→P4 and P1→P3→P4). We have added clarification to this effect in the legend of Figure 7 (Main manuscript Page 14).

Reviewer #3:

I would like to thank the authors for their work in submitting a new improved version of the manuscript.

Thank you for performing the permutation test. However, there are still elements in this procedure that are unsatisfactory. Most importantly, the number of permutations, 50, is too small. Indeed, a p-value is the probability of an extreme event and thus requires more repetitions. I would suggest 1000 repetitions.

Also, the text is too vague to understand how is the procedure done. I read “To do so, we utilized the whole discovery set (i.e., 297 CN and 602 MCI/Dementia participants from ADNI2/GO) and randomly shuffled the labels of ‘CN’ and ‘PT’ to create a null dataset with 297 new ‘CN’ and 602 ‘PT’ whose distributions are comparable.”

Please be specific on the method used. “Whose distribution is comparable” is vague. Next, it would be helpful to know what exactly is inside the permutation loop.

A description similar to Algorithm 1 would be helpful.

Also, I would recommend redoing this for various values of M, say, M=2,3,4,5.

Indeed, the number of clusters is a well-known difficult parameter to estimate and is critical in the manuscript.

Response: We want to thank the reviewer for raising these important issues. We only ran 50 permutations in previous revision because of the computational burden of training a GAN model. However, from a statistical perspective, we agree that a larger number of repetitions is needed to draw a statistical conclusion with relative high confidence. Leveraging parallel computing and additional computational power, we have performed 1000 permutations for M=2-5 with a smaller number of holdout cross-validation folds. This more rigorous permutation test also shows significance of reproducibility of the smile-GAN patterns for M=3-5 (Supplemental Section 2.4). Results with M=2 also show significant reproducibility. The null ARIs with M=2 have a mean of 0.052 and standard deviation of 0.049, indicating the significance of the observed ARI=0.67 ($p < 0.001$). However, we did not include the test result for M=2 in the supplemental material because M=2 generally stratified the data into mild and severe atrophy patterns, which is not clinically interesting as described in Method section 4.6.

We apologize for not explaining the permutation procedure clearly. By ‘whose distribution is comparable’, we mean that, after random shuffling of diagnosis labels, the new ‘CN’ and ‘PT’ group, sampled from same distribution, have comparable distributions in ROI volumes (input for Smile-GAN model). We have rephrased the sentence to make it clearer. More importantly, we have added an Algorithm 2 (Supplemental Page 7) to give more details of the permutation loop and hold-out cross validation procedure for deriving ARIs inside each permutation loop. Detailed results and additional descriptions of the permutation process were included in the Supplemental section 1.4/2.4 and Algorithm 2 (Supplemental Page 7 and 13). The Permutation Loop is included here:

Permutation Experiment Procedure

for i from 1 to 1000 do

Randomly shuffle diagnosis labels of all 899 participants;

Select out the new ‘CN’ group and ‘PT’ group;

for j from 1 to 10 do

*Randomly sample 80% of participants from 'CN' and 'PT' group respectively
Run Smile-GAN model with 80% sampled 'PT' and 'CN' data and derive cluster labels
for all 'PT' data;*

end

*Calculate pairwise ARIs among 10 lists of derived labels and saved the mean value as
the ARI for the i th permutation;*

end

Next, for improving readability, could you define the use of the word “Domain” in Fig 8? Would a simpler word like vector or long vector suffice? Is the dimension the number of ROI? Is it 145 for all experiments?

Response: We thank the reviewer for pointing out these points of ambiguity. The word ‘Domain’ refers to the set of all possible data. For example, CN domain represents the set of all CN data. We have added additional explanation for PT domain and CN domain in the main manuscript (Main section 4.1 on Page 18). The dimension of features of CN domain and PT domain are both 145 for this task and we used 145 ROIs for all experiments described in the main manuscript and supplemental material. However, the number and type of data points in each domain (set) can be varied; this figure aims to demonstrate a more generic formulation of the Smile-GAN methodology than the ROI-based implementation used in this manuscript.

Also, “atrophy” is used extensively. Could it be defined early? Similarly for the ROIs. Are the white matter regions excluded? what about the ventricles?

Response: We have added a definition of ‘atrophy’ and some more information on the nature of the ROIs in the second paragraph of the introduction (Page 3):

“...specifically heterogeneity of atrophy as measured by decreases in volumes of gray matter and white matter regions of interest and increases in ventricle volumes...”

Additional details on ROIs are provided in Methods section 4.3 (Page 20):

“...145 anatomical regions of interest (ROIs) were identified in gray matter (GM, 119 ROIs), white matter (WM, 20 ROIs) and ventricles (6 ROIs) using a multi-atlas label fusion method.” While both white matter regions and ventricles are included in 145 ROIs for model training, in VBM results we only show atrophy in gray matter to display the most distinct patterns. In Semi-synthetic Test, we only introduced atrophy to selected gray matter regions (decrease volumes of selected GM regions) without changing white matter and ventricle volumes. We have made these points clearer in Method section 4.3 (Page 20) and Supplemental Method 1.3.2 (Supplemental Page 5).

REVIEWER COMMENTS

Reviewer #1 (Remarks to the Author):

I am happy with the responses from the authors and changes made to the manuscript and have no further comments.

Reviewer #3 (Remarks to the Author):

Thank you for this second revision of the manuscript.

I do have 2 remaining comments

1) In algorithm 2, it is not explicit how you computed the ARI corresponding to the true CN/PT data i.e. not shuffled. I expect that you ran the same Algorithm 2, except that you did not shuffle the data. Is it the case?

2) I understand your explanation about the word domain. However, it seems almost similar to the word data. In order to reduce the use of jargon, could you just use data?

Dear Editor and Reviewers:

We are grateful to the reviewers for their constructive comments, which have been very helpful in revising our manuscript and in improving its quality.

We addressed the questions raised by the reviewers as follows. The primary changes and corrections are marked in red in the revised manuscript.

Reviewer #3:

Thank you for this second revision of the manuscript.

I do have 2 remaining comments

1) In algorithm 2, it is not explicit how you computed the ARI corresponding to the true CN/PT data i.e. not shuffled. I expect that you ran the same Algorithm 2, except that you did not shuffle the data. Is it the case?

Response: We want to thank the reviewer for pointing out this ambiguous point. It is correct that we ran the same inner loop of algorithm 2 without shuffling the data to derive observed ARI. We have added one sentence to explain that in supplemental section 1.4. Detailed descriptions on deriving observed ARIs can be also found in Main section 4.6.

2) I understand your explanation about the word domain. However, it seems almost similar to the word data. In order to reduce the use of jargon, could you just use data?

Response: We agree that the word 'domain' may lead to confusions, and we think that 'group' might be a more suitable word to replace it. We have changed the word 'domain' in both texts and figures.

REVIEWERS' COMMENTS

Reviewer #3 (Remarks to the Author):

I am satisfied with the answers of the authors. I have no further remarks at this point.